

# Machine dependence as a source of uncertainty in climate models: The HadGEM3-GC3.1 CMIP Preindustrial simulation

Maria-Vittoria Guarino[1], Louise C. Sime[1], David Schroeder[2], Grenville M. S. Lister[3], and
Rosalyn Hatcher[3]

[1]British Antarctic Survey, Cambridge, UK
[2]Department of Meteorology, University of Reading, Reading, UK
[3]National Centre for Atmospheric Science, University of Reading, Reading, UK

**Correspondence:** Maria-Vittoria Guarino (m.v.guarino@bas.ac.uk)

**Abstract.**

When the same weather or climate simulation is run on different High Performance Computing (HPC) platforms, model outputs may not be identical for a given initial condition. While the role of HPC platforms in delivering better climate projections is often discussed in literature, attention is mainly focused on scalability and performance rather than on the impact

of machine-dependent processes on the numerical solution. At the same time, machine dependence is an overlooked source of uncertainty when it comes to discussing the model spread observed within the Coupled Model Intercomparison Projects (CMIP).

Here we investigate the impact of machine dependence on model results and quantify, for a selected case study, the magnitude of the uncertainty. We consider the Preindustrial (PI) simulation prepared by the UK Met Office for the forthcoming CMIP6.

We compare key climate variables between PI control simulations run on the UK Met Office supercomputer and the ARCHER HPC platform. Discrepancies strongly depend on the timescale. Decadal means show substantial differences of up to 0.2 °C for global mean air temperature, 1 W /m$^2$ for TOA outgoing longwave flux and 1.2 million km$^2$ for Southern Hemisphere sea ice area. However, on multi-centennial timescales the differences are not significant and the long-term statistics of the two runs are similar.

Differences between the two simulations can be linked to variations in the strongest modes of climate variability. In the Southern Hemisphere, this results in large SST anomalies where ENSO teleconnection patterns are expected that can reach 0.6 °C (and SNR ≥ 1) even on centennial timescales.

## 1   Introduction

Coupled model intercomparison projects (CMIP) have become an indispensable tool to progress our understanding of the Earth
system and inform decision makers on the state of the climate.

The latest Coupled Model Intercomparison Project Phase 5 (CMIP5) collected numerical simulations performed with 60 models by 26 research institutes around the world. The follow-on CMIP6 archive, to be completed by December 2020, is expected to gather model outputs from 32 research institutes.



CMIP models differ among each other in their physical formulation, numerical discretization, physical process parametrizations and code implementation. As each model rightfully represents a possible realization of the real world, it has been long recognized that multi-model means are more meaningful than single-model means for any variable. The spread (standard deviation) among models associated to multi-model means is usually explained as the combination of model uncertainty (which accounts for all the aforementioned inter-model differences) and internal (natural) climate variability, a consequence of the chaotic nature of the climate system. Furthermore, for future climate projections, the uncertainty associated with possible future greenhouse gas emissions is classified separately as 'scenario uncertainty' (Hawkins and Sutton (2009) and Hawkins and Sutton (2011)).

Recently, a few studies emphasized the need of separating the sources of uncertainty in climate simulations to learn about their relative importance (e.g., Cox and Stephenson (2007), Knutti et al. (2008)), finding a robust way of estimating the internal climate variability (needed for detection and attribution studies) (e.g., Kay et al. (2015), Olonscheck and Notz (2017)), and reducing the overall uncertainty of climate projections (e.g., Hawkins and Sutton (2009) and Hawkins and Sutton (2011)).

All these studies overlook the existence of another source of uncertainty associated with the computing environment where numerical simulations are run. This type of uncertainty is due to machine-dependent processes and can contribute to the model spread observed within the CMIP archives, and any other model intercomparison project, by an amount that has never been quantified.

The issue of being able to reproduce identical simulation results across different supercomputers, or following a system upgrade on the same supercomputer, has long been known by numerical modellers and computer scientists. However, the impact that a different computing environment can have on otherwise identical numerical simulations appears to be little known by climate models users and model data analysts. In fact, the subject has never received much attention. It sometimes occupies a few sentences in research papers (see for example Lunt et al. (2012)) or is completely ignored.

While one could argue that 'machine dependence uncertainty' is embodied in the model uncertainty component we recall that, being based on different physical and numerical formulations, model uncertainty has a completely different nature. Additionally, independent source codes and physics implementations have to be encouraged among the community to preserve the independence across numerical experiments (see Abramowitz et al. (2019) for a review on the topic). On the contrary, machine dependence is a component of the total uncertainty of climate simulations which has the potential to be reduced or completely removed (if, for example, all the models within the same model intercomparison project were run on the same supercomputer).

To the extent of our knowledge, only a few authors discussed the existence of machine dependence uncertainty and highlighted the importance of bit-for-bit numerical reproducibility in the context of climate model simulations. Song et al. (2012) and Hong et al. (2013) investigated the uncertainty due to the round-off error in climate simulations. Liu et al. (2015b) and Liu et al. (2015a) discussed the importance of bitwise identical reproducibility in climate models, highlighting the lack of attention that the climate modelling community gives to the subject.

In this paper, we investigate the machine dependence of climate model simulations by studying the behaviour of the UK CMIP6 PI control simulation with the HadGEM3-GC3.1 model on two different HPC platforms. We quantify the magnitude of the uncertainty attributable to machine-dependent processes and its impact on the physical interpretation of model outputs.



Note that the PI control simulation is a constant-forcing experiment and serves the purpose of estimating the internal climate variability. No ensemble members are run for such experiment because, provided that the simulation is long enough, this will return a picture of the natural climate variability.

We will focus on estimating how long constant-forcing climate simulations should be for machine dependence uncertainty to become negligible. In the paper, discrepancies between the means of key climate variables like SST, Sea Ice Concentration, 2m Air Temperature, LW and SW TOA radiation fluxes and Precipitation flux will be analysed at different timescales, from decadal to centennial, and linked to variations in the strongest modes of climate variability.

The remainder of the paper is organized as follows. In section 2, mechanisms by which the computing environment can influence the numerical solution of chaotic dynamical systems are reviewed and discussed. In section 3, the numerical simulations are presented and the methodology used for the data analysis is described. In section 4, the simulation results are presented and discussed. In section 5, the main conclusions of the present study are summarized.

## 2   The impact of machine dependence on the numerical solution

In this section, possible known ways in which machine-dependent processes can influence the numerical solution of chaotic dynamical systems are reviewed and discussed.

Different compiling options, degrees of code optimization and basic library functions all have the potential to affect the reproducibility of model results across different HPC platforms, and on the same platform under different computing environments. Here we provide a few examples of machine-dependent numerical solutions using the 3D Lorenz model (Lorenz, 1963), which is a simplified model for convection in deterministic flows. The Lorenz model consists of the following three differential equations:

$$
\begin{aligned}
\frac{dx}{dt} &= \alpha(y - x) \\
\frac{dy}{dt} &= \gamma x - y - zx \\
\frac{dz}{dt} &= xy - \beta z
\end{aligned}
\tag{1}
$$

where the parameters $\alpha = 10$, $\gamma = 28$ and $\beta = 8/3$ were chosen to allow the generation of flow instabilities and obtain chaotic solutions (Lorenz, 1963). The model was initialized with $(x_0, y_0, z_0) \equiv (1, 1, 1)$ and numerically integrated with a 4th-order Runge-Kutta scheme using a time step of 0.01. The Lorenz model was run on two HPC platforms, namely: the UK Met Office Supercomputer (hereinafter simply "MO") and ARCHER.

To demonstrate first the implications of switching between different computing environments, the Lorenz model was run on the ARCHER platform using:

- two different FORTRAN compilers (cce8.5.8 and intel17.0), see Figure 1a and 1b;

- same FORTRAN compiler (cce8.5.8) but different degrees of floating-point optimization (-hfp0 and -hfp3), see Figure 1c and 1d;





– same FORTRAN compiler and compiling options but the x-component in (1) was perturbed by adding a noise term obtained using the `random_number` and `random_seed` intrinsic FORTRAN functions. In particular, the seed of the random number generator was set to 1 and 3 in two separate experiments, see Figure 1e and 1f.

Finally, to illustrate the role of using different HPC platforms, the Lorenz model was run on the ARCHER and MO platforms
using the same compiler (intel17.0) and identical compiling options (i.e. level of code optimization, floating-point precision, vectorization) (Figure 1g and 1h).

The divergence of the solutions in Figure 1a and 1b can likely be explained by the different 'computation order' of the two compilers (i.e. the order in which a same arithmetic expression is computed). In Figure 1c and 1d, solutions differ because of the round-off error introduced by the different precision of floating-point computation. In Figure 1e and 1f, the different seed used
to generate random numbers caused the system to be perturbed differently in the two cases. While this conclusion is straightforward, it is worth mentioning that the use of random numbers is widespread in weather and climate modelling. Random number generators are largely used in physics parametrizations for initialization and perturbation purposes (e.g. clouds, radiation and turbulence parametrizations) and, as obvious, in stochastic parametrizations. The processes by which initial seeds are selected within the model code are thus crucial in order to assure numerical reproducibility. Furthermore, different compilers may have
different default seeds.

As for Figure 1g and 1h, this is probably the most relevant result for the present paper. It highlights the influence of the HPC platform (and of its hardware specifications) on the final numerical solution. In Figure 1g and 1h the two solutions diverge in time similarly to Figure 1a - 1d, however identifying reasons for the observed differences is not straightforward. While we speculate that reasons may be down to machine architecture and/or chip-set, further investigations on the subject were not
pursued as this would be beyond the scope of this study.

The three mechanisms discussed above were selected because illustrative of the problem and easily testable via a simple model such as the Lorenz model. However, there are a number of additional software and hardware specifications that can influence numerical reproducibility, and that only emerge when more complex codes, like weather and climate models, are run. These are: number of processors and processor decomposition, communications software (i.e. MPI libraries), threading (i.e.
OpenMP libraries).

We conclude this section stressing that the four case studies presented in Figure 1 (and the additional mechanisms discussed in this section) are all essentially a consequence of the chaotic nature of the system. When machine-dependent processes introduce a small perturbation/error into the system (no matter by which mean), they cause it to evolve differently after a few time-steps.





## 3 Methodology

### 3.1 Numerical simulations

In this study, we consider two versions of the Preindustrial (PI) control simulation prepared by the UK Met Office for the sixth coupled model intercomparison project (CMIP6) (Eyring et al., 2016). This PI control experiment is used to study the

(natural) unforced variability of the climate system and it is one of the reference simulations against which all the other CMIP6 experiments will be analysed.

The PI simulation considered in this paper uses the N96 resolution version of the HadGEM3-GC3.1 climate model (N96ORCA1). The model set-up, initialization, performance and physical basis are documented in Menary et al. (2018) and Williams et al. (2018), to which publications the reader is referred for a detailed description. In summary, HadGEM3-GC3.1 is a global cou-

pled atmosphere-land-ocean-ice model that comprises the Unified Model (UM) atmosphere model (Walters et al., 2017), the JULES land surface model (Walters et al., 2017), the NEMO ocean model (Madec et al., 2015) and the CICE sea ice model (Ridley et al., 2018). The UM vertical grid contains 85 pressure levels (terrain-following hybrid height coordinates) while the NEMO vertical gird contains 75 depth levels (rescaled-height coordinates). In the N96 resolution version, the atmospheric model utilizes a horizontal grid-spacing of approximately 135 km on a regular latitude-longitude grid. The grid spacing of the

ocean model, which employs an ortoghonal curvilinear grid, is $1°$ everywhere but decreases down to $0.33°$ between $15°$ N and $15°$ S of the equator, as described by Kuhlbrodt et al. (2018).

Following the CMIP6 guidelines, the model was initialized using constant 1850 GHGs, ozone, solar, tropospheric aerosol, stratospheric volcanic aerosol and land use forcings. The UK CMIP6 PI control simulation (hereinafter referred to as $PI_{MO}$) was originally run on the MO HPC platform on 2500 cores. The model was at first run for 700 model-years to allow the

atmospheric and oceanic masses to attain a steady state (model spin-up), and then run for further 500 model-years (actual run length) (see Menary et al. (2018) for details). A copy of the PI control simulation was ported to the ARCHER HPC platform (hereinafter referred to as $PI_{AR}$), initialized using the atmospheric and oceanic fields from the end of the spin-up and run for 200 model-years using 1500 cores. The source codes of the atmosphere and ocean models were compiled on the two platforms using the same levels of code optimization (`-O` option), vectorization (`-Ovector` option), floating-point precision

(`-hfp` option) and, for numerical reproducibility purposes, selecting the least tolerant behaviour in terms of code optimization when the number of ranks or threads varies (`-hflex_mp` option). For the atmosphere component the following options were used: `-O2 -Ovector1 -hfp0 -hflex_mp=strict`. For the ocean component the following options were used: `-O3 -Ovector1 -hfp0 -hflex_mp=strict` .

Table 1 provides an overview of the hardware and software specifications of the two HPC platforms where the model was

run.

Of the possible mechanisms discussed in section 2, the ARCHER and MO simulations were likely affected by differences in compiler, processor type, number of processors and processor decomposition (alongside the different machine).



**Table 1.** Hardware and software specifications of the ARCHER and MO HPC platforms as used to run the HadGEM3-GC3.1 model.

| HPC Platform | Machine | Compiler | Processor |
|---|---|---|---|
| MO | Cray XC40 | cce 8.3.4 | Broadwell |
| ARCHER | Cray XC30 | cce 8.5.5 | Ivy Bridge |

## 3.2 Data post-processing and analysis

During the analysis of the results, the following climate variables were considered: sea surface temperature (SST), sea ice area/concentration (SIA/SIC), 1.5m air temperature (SAT), the outgoing long-wave and short-wave radiation fluxes at top of the atmosphere (LW TOA and SW TOA), and the precipitation flux (P). These variables were selected as representative of the
ocean and atmosphere domains and because they are commonly used to evaluate the status of the climate system.

Discrepancies between the means of the selected variables were analysed at different timescales, from decadal to centennial. To compute 10-, 30-, 50- and 100-year means, (PI$_{MO}$ - PI$_{AR}$) 200-year time-series were divided into 20, 6, 4 and 2 segments respectively. Spatial maps were simply created by averaging each segment over time. Additionally, to create the scatter plots presented in section 4.1, the time average was combined with an area-weighted spatial average. Except for SIC, all the
variables were averaged globally. Additionally, SIC, SST and SAT were regionally-averaged over the Northern and Southern Hemisphere, while SW TOA, LW TOA and P were regionally-averaged over the tropics, Northern extra-tropics and Southern extra-tropics according to the underlying physical processes.

Note that, when calculating (PI$_{MO}$ - PI$_{AR}$) differences, PI$_{MO}$ and PI$_{AR}$ segments are subtracted in chronological order. Thus, for example, the first 10 years of PI$_{AR}$ are subtracted from the first 10 years of PI$_{MO}$ and so on. In fact, because the PI
control simulation is run with a constant climate forcing, using a 'chronological order' in the strictest sense is meaningless, as every 10 years segment is equally representative of the pre-industrial decadal variability. We acknowledge that an alternative approach, equally valid, would be to subtract PI$_{AR}$ and PI$_{MO}$ segments without a prescribed order. This approach would probably lead to a spread between the simulations not identical to the one presented in section 4.1, but not more (or less) meaningful. More importantly, by following a chronological order, we compare same time-steps. This is helpful in investigating
the impact of machine dependence on model results.

The divergence of the results between the two runs was quantified by computing the Signal-to-Noise Ratio (SNR) for each considered variable at each timescale. The signal is represented by the mean of the differences between PI$_{MO}$ and PI$_{AR}$ ( $\mu_{MO-AR}$ ) and the noise is represented by the standard deviation of (PI$_{MO}$ - PI$_{AR}$) ( $\sigma_{MO-AR}$ ) divided by $\sqrt{2}$ (see below for details). Thus, SNR is defined as:

$$SNR = \frac{|\mu_{MO-AR}|}{\frac{|\sigma_{MO-AR}|}{\sqrt{2}}} \qquad (2)$$




when SNR < 1, ($PI_{MO}$ - $PI_{AR}$) differences can be interpreted as fluctuations of the system not necessarily linked to machine dependence (i.e. $PI_{MO}$ and $PI_{AR}$ do not differ more than $PI_{MO}$ (or $PI_{AR}$) evaluated at two different points in time). When SNR > 1, the observed differences are outside of the expected range of variability and $PI_{MO}$ and $PI_{AR}$ are considered to be different.

Note that (2) makes use of two mathematical assumptions: $PI_{MO}$ and $PI_{AR}$ are uncorrelated (i.e. their covariance is zero), and have the same variance. Under these assumptions, the noise can be represented as the standard deviation of the differences between $PI_{MO}$ and $PI_{AR}$ divided by $\sqrt{2}$.

While the first assumption is justified by the chaotic nature of the system, which causes the two simulations to start diverging after the first few time-steps, the equal variances assumption only holds when variables are analysed on a 200-year timescale.

This assumption weakens and breaks down at smaller timescales, where ($PI_{MO}$ - $PI_{AR}$) differences are the largest. While one could think to use $\sigma_{AR}$ and/or $\sigma_{MO}$ in (2) to evaluate SNR on shorter timescales, we point out that a change of methodology is not necessary. In fact, by dividing ($\mu_{MO-AR}$) by ($\sigma_{MO-AR}/\sqrt{2}$) in (2) we decrease the signal-to-noise ratio, as the standard deviation of ($PI_{MO}$ - $PI_{AR}$) differences is higher than the standard deviation of the single $PI_{MO}$ and $PI_{AR}$ time-series. Thus, on short timescales, a SNR > 1 computed using (2) would only be larger if the above alternative method were to be used.

For the final step of the analysis, the El Niño Southern Hemisphere Oscillation (ENSO), the Southern Annular Mode (SAM) and the North Atlantic Oscillation (NAO) indexes were computed for the ARCHER and MO simulations. These three climate oscillations help us investigate the behaviour of the climate system across the simulations. We chose the NINO3.4 index with a 3-month running mean to represent the ENSO signal (Trenberth, 1997), the Gong and Wang index (Gong and Wang, 1999) based on annual data to represent the SAM signal, and the winter Hurrell index (Hurrell, 1995) to represent the NAO signal.

The indexes are defined as follows:

$$NINO3.4 = SST_{mnth} - \overline{SST_{30yr}} \quad \text{if} \quad 5° \, N \leq \text{latitude} \leq 5° \, S \quad \text{and} \quad 120° \, W \leq \text{longitude} \leq 170° \, W \tag{3}$$

$$SAM \ index = SLP^*_{40°S} - SLP^*_{65°S} \quad \text{where} \quad SLP^* = \frac{SLP_{mnth} - \overline{SLP_{30yr}}}{STDEV\left(SLP_{30yr}\right)} \tag{4}$$

$$NAO \ index = SLP^*_{Lisbon} - SLP^*_{Reykjavik} \quad \text{where} \quad SLP^* = \frac{SLP_{DJFM} - \overline{SLP_{longterm}}}{STDEV\left(SLP_{longterm}\right)} \tag{5}$$

where $SST_{mnth}$ and $SLP_{mnth}$ are monthly sea surface temperature and sea level pressure (SLP) values and $SLP_{DJFM}$ are

SLP December-January-February-March seasonal means. For the ENSO and SAM indexes, the climatological means of the first 30 years of simulation ($\overline{SST_{30yr}}$ and $\overline{SLP_{30yr}}$) were used to compute the anomalies. For the NAO index, SLP anomalies were computed using the long-term mean ($\overline{SLP_{longterm}}$). Finally, note that $SLP^*_{40°S}$ and $SLP^*_{65°S}$ are zonal means, while $SLP^*_{Lisbon}$ and $SLP^*_{Reykjavik}$ correspond to single-point SLP values over Lisbon (PT) and Reykjavik (IS) obtained through interpolation on the model grid.



## 4    Results and discussion

### 4.1    Timescales of divergence

The long-term means of the selected variables, and the associated SNR, are shown in Figures 2 and 3. All the variables exhibit a SNR < 1, indicating that on multi-centennial timescales the differences observed between the two simulations fall into the expected range of variability of the PI control run. However, it is worth noting that some variables have a SNR close to 1 in both the Southern and Northern Hemisphere; see for example SST in Figure 2b, where SNR ≈ 0.9, or SIC in Figure 2d and 2h, where SNR ≈ 0.7. The strengthening of the anomalies is geographically confined to areas where the ENSO and NAO climate modes primarily manifest, a physical interpretation of the enhancement of SNR in these areas will be provided in section 4.2.

When maps like the ones in Figure 2 and 3 are computed using 10-, 30-, 50- and 100-year averaging periods (not shown), the magnitude of the anomalies increase and ($PI_{MO}$ - $PI_{AR}$) differences become significant (SNR » 1). This behaviour is discussed below.

Figures 4 to 9 show annual-mean time-series of spatially averaged SST, SIA, SAT, SW TOA, LW TOA and P, respectively. Figures 4d to 9d show ($PI_{MO}$ - $PI_{AR}$) differences as a function of the averaging timescale for each variable (see section 3.2 for details on the computation of the means). The 200-year global-mean and standard deviation of each variable are shown in Table 2.

For all the considered variables, $PI_{MO}$ and $PI_{AR}$ start diverging quickly after the first $1 - 2$ years of simulation, once the system has lost memory of the initial conditions. See section 2 (Figure 1) for further discussion on how machine-dependent processes can influence the temporal evolution of the system.

SST, SAT, SW TOA and LW TOA differ the most in the Northern Hemisphere (and particularly on decadal timescales) (yellow diamonds in Figures 4d,6d,7d,8d), while SIA anomalies are particularly high in the Southern Hemisphere (red crosses in Figure 5d) and P anomalies in the tropics (green circles in Figure 9d). Overall, discrepancies are the largest at decadal timescales where the spread between the two simulations can reach |0.2| °C in global mean air temperature (Figure 6d), |1.2| million km$^2$ in Southern Hemisphere sea ice area (Figure 5d), or |1| W /m$^2$ in global TOA outgoing LW flux (Figure 8d).

As the timescale increases, ($PI_{MO}$ - $PI_{AR}$) differences get smaller and approach zero when a 200-year timescale is considered. This happens because 200 years is a long enough averaging-period for the positive and negative extremes in the time-series of Figure 4 - 9 to average out. On shorter time-intervals, strong increasing/decreasing trends in one simulation may not be compensated by trends of opposite sign in the the other simulation and may result in SNR » 1. See for example the first 10 years of the NH SIA time-series in Figure 5a. Additionally, the 200-year mean of the SIA seasonal cycle shown in Figure 5c is almost identical for ARCHER and MO, confirming that on a 200-year timescale the two runs are comparable. This suggests that the overall physical behaviour of the model has not been affected by the porting.

It is worth noting that 200 years is a considerable length for a fully coupled (Atmosphere–Ocean–Sea Ice–Land Surface) climate model simulation. A 200-year run is not always possible because of the significant computational resources required. As an example, the CMIP6 minimum run length requirement for most of the Model Intercomparison Projects (MIP) is 100 years. Our results suggest that 100 years may not be enough to make machine dependence influences negligible. This is particularly





**Table 2.** 200-year global mean and standard deviation for SST, SIA, SAT, SW TOA, LW TOA and P.

|  | MO | ARCHER |
|---|---|---|
|  | Mean , StDev | Mean , StDev |
| SST (°C) | 17.93 , 0.07 | 17.95 , 0.08 |
| SIA ($10^6$ km$^2$) | 21.44 , 0.65 | 21.30 , 0.68 |
| SAT (°C) | 13.71 , 0.10 | 13.75 , 0.12 |
| SW TOA (W /m$^2$) | 98.83 , 0.24 | 98.76 , 0.27 |
| LW TOA (W /m$^2$) | 241.29 , 0.27 | 241.36 , 0.33 |
| P ($10^{-6}$ kg /m$^2$ /s) | 36.22 , 0.12 | 36.25 , 0.14 |

true when we look at the spatial patterns of ($PI_{MO}$ - $PI_{AR}$) differences (see section 4.2 for further discussion). Additionally, when model intercomparison analyses are conducted on climatological (30 years) or decadal (10 years) timescales, results in Figures 4-9 indicate that the multiplicity of HPC platforms used to perform the simulations may substantially contribute to the spread among models.

5    In Figures 4d to 9d, the variation of ($PI_{MO}$ - $PI_{AR}$) with the timescale suggests the existence of power law relationship[1]. To investigate this behaviour, a base-10 logarithmic transformation was applied to the x- and y-axes of Figure 4d to 9d and linear regression was used to find the straight-lines that best fit the data.

Figure 10 shows log-log plots for SST, SAT, SW TOA, LW TOA and P for the maximum ($PI_{MO}$ - $PI_{AR}$) values at each timescale. To ease the comparison, all the variables were averaged globally and over the SH and NH Hemispheres. Global,

10   NH and SH mean data all align along a straight line, confirming the existence of a power law. However, the most interesting result emerges at the global scale where ($PI_{MO}$ - $PI_{AR}$) differences vary following a same power law relationship, regardless the physical quantity considered. More precisely, the actual slope values for SST, SAT, SW TOA, LW TOA and P are: -0.65, -0.65, -0.64, -0.66, -0.67 respectively. Thus, all the straight-lines that best fit the global mean data in Figure 10 have a slope of $\approx$ 2/3. The existence of a $\approx$ 2/3 power law, which does not depend on the single quantity, suggests that the machine-induced

15   uncertainty scales consistently with the timescale across the whole climate simulation.

SIA (not shown) was the only variables that did not show a $\approx$ 2/3 power law relationship. This however should not invalidate the analysis presented above. The sea ice area is an integral computed on a limited area, and not a mean computed on a globally uniform surface (like all the other variables considered here), and thus represents a signal of a different nature.

---

[1]Note that, for readability, the ticks of the x-axes of Figures 4d to 9d were equally spaced. This partially masks the power law behaviour discussed in the paper, which can be better detected when the natural x-axes are used.





## 4.2 The physical implications

In this section, the impact of machine dependence on model results is analysed from a physical point of view. As mentioned in section 4.1, ($PI_{MO}$ - $PI_{AR}$) differences materialize into spatial patterns that are signatures of physical processes (Figure 2 and 3). The interpretation of such patterns can sometimes be misleading. For example, while Figure 3g may suggest a westward

shift of the Intratropical Convergence Zone (ITCZ) between $PI_{MO}$ and $PI_{AR}$, the associated SNR in Figure 3h is small ($\leqslant 0.4$) and indicates that the two simulations cannot be considered significantly different on a 200-year timescale.

However, unlike P, other variables like SST and SIC have a high SNR despite the use of a 200-year averaging period. Interestingly, for these variables, SNR is maximum where the ENSO and NAO climate modes subsist and it becomes larger than one on centennial scales (Figure 11).

To assess whether the climate system of the ARCHER and MO simulations behaves differently, the ENSO (Figure 12a), NAO (Figure 12b) and SAM (Figure 12c) indexes were computed for $PI_{MO}$ and $PI_{AR}$ as described in section 3.2. These three climate oscillations were selected because they interact with most of the physical processes governing the climate system. While the NAO and SAM signals are presented in Figure 12 for completeness, the following analysis will mainly focus on the ENSO signal, as an example of the impact of machine dependence on model results.

The connection between SIC (and SST) anomalies in the Southern Hemisphere and ENSO has been widely documented in literature, e.g. Kwok and Comiso (2002), Liu et al. (2002), Turner (2004), Welhouse et al. (2016), Pope et al. (2017).

The warm (El Niño) and cold (La Niña) phases of ENSO manifest in the Amundsen–Bellingshausen Sea (ABS) and Ross Sea sectors in a diametrically opposite way. While El Niño events favour an increase of winter sea ice concentration in the ABS sector and a decrease of winter sea ice concentration in the Ross Sea sector, La Niña events are associated with negative SIC

anomalies in the ABS sector and positive SIC anomalies in the Ross Sea sector (Kwok and Comiso (2002), Pope et al. (2017)).

In Figure 2g and 2h, SIC anomalies and the associated SNR are the largest in West Antarctica where ENSO teleconnection patterns are expected. This suggests that ($PI_{MO}$ - $PI_{AR}$) differences are driven by two different ENSO regimes.

This hypothesis is confirmed by the ENSO signal in Figure 12a. A few times, to a strong El Niño (/La Niña) event in $PI_{MO}$ corresponds a strong La Niña (/El Niño) event in $PI_{AR}$. This opposite behaviour enlarges SIC (and SST) differences between

the two runs and strengthens the $\mu_{MO-AR}$ signal, resulting in a strong SNR. Additionally, $PI_{MO}$ have a few more EL Niño events (orange shaded areas in Figure 12a) than $PI_{AR}$. While in Figure 12a the ENSO index of $PI_{MO}$ is greater than/or equal to 0.5 (threshold value for the ENSO onset) 894 times, the ENSO index of $PI_{AR}$ is greater than/or equal to 0.5 768 times. At the same time, $PI_{AR}$ seems to simulate overall more intense La Niña events (green shaded areas in Figure 12a), as its ENSO index takes values in the interval [-2 , -3] more often than $PI_{MO}$. This analysis is supported by the positive 200-year mean

SIC anomalies in the ABS sector in Figure 2g (i.e. more El Niño events result in larger $PI_{MO}$ sea ice concentration) and the corresponding negative SST anomalies in Figure 2c.

Finally, because of the same mechanism described above, SST and SIC exhibit a SNR larger than one when a 100-year timescale is used (Figure 11). This result highlights that, even on a climate (/long) timescale, the uncertainty introduced by machine dependence may be not negligible.





## 5 Conclusions

In this paper, machine dependence is discussed. Two versions of the UK CMIP6 PI control simulation run on the UK Met Office supercomputer (MO) ($\text{PI}_{MO}$) and ARCHER ($\text{PI}_{AR}$) HPC platforms were used to illustrate the impact of machine dependence of coupled climate model simulations. Simulations used the N96ORCA1 HadGEM3-GC3.1 model. Discrepancies between the

means of key climate variables (SST, SIA/SIC, SAT, SW TOA, LW TOA and P) were analysed at different timescales, from decadal to centennial.

Although the two versions of the same PI control simulation do not bit-compare, we found that the long-term statistics of the two runs are similar and that, on multi-centennial timescales, the considered variables show a signal-to-noise ratio (SNR) less than one. However, inconsistencies between the two runs increase and become significant (SNR $\geq$ 1) for shorter timescales,

being the largest at decadal timescales. For example, when a 10-year averaging period is used, machine dependence can account for up to $|0.2|$ °C global mean air temperature anomalies, or $|1.2|$ million km$^2$ Southern Hemisphere sea ice area anomalies.

Differences between the two simulations can be linked to variations in the strongest modes of climate variability. In the Southern Hemisphere, this results in large SST anomalies where ENSO teleconnection patterns are expected that can reach 0.6 °C (and SNR 1) even on centennial timescales.

The relationship between global mean differences and timescale exhibits a 2/3 power law behaviour, regardless the physical quantity considered. This suggests a consistent time-dependent scaling of the machine-induced bias across the whole climate simulation.

CMIP6 guidelines recommend a minimum simulation length of 100 years for most of the MIP experiments. Our results suggest that 100 years may not be enough to make machine dependence influences negligible. Additionally, Figure 4d-9d

indicate that model intercomparison analyses conducted on climatological (30 years) or decadal (10 years) timescales may be substantially influenced by the diversity of HPC platforms used to perform the simulations. This result is in contrast with what previously found by Song et al. (2012), who stated that for (30-year) climatological means machine dependence uncertainty is negligible.

Because machine dependence uncertainty is essentially a consequence of the sensitivity of the climate system to the ini-

tial conditions (see section 2 for a more detailed discussion), repeating a similar analysis but using varying-forcing climate simulations and ensemble means would be a straightforward extension of this work. The question to answer is whether the spread between the same set of ensemble members run on the ARCHER and MO platforms is any different. Song et al. (2012) addressed the same question (using the Community Climate System Model Version 3) showing that a minimum of 15 ensemble members are needed to make machine dependence uncertainty negligible.

CMIP6 guidelines advise modelling groups to create ensembles for the so-called 'historical simulations' with a minimum of 3 members. The small number of ensemble members required, unlikely to be exceeded by much by modelling groups because of computational resources availability reasons, is a further indication that machine dependence may contribute to the spread among CMIP models even when ensemble means for each model are considered.




While the quantitative analysis presented in this paper applies strictly to HadGEM3-GC3.1 constant-forcing climate simulations only, this study has the broader purpose of increasing the awareness of the climate modelling community on the subject of machine dependence of climate simulations.

The results presented here have immediate applications for those members of the the UK CMIP6 community who will run
individual MIP experiments on the ARCHER HPC platform, and will compare their results against the reference PI simulation run on the MO platform by the UK Met Office. In particular, the magnitude of $(\text{PI}_{MO} - \text{PI}_{AR})$ differences presented in this paper should be regarded as a threshold value below which differences between ARCHER and MO simulations must be treated as suspicious. Although our results are based on a single case study, they suggest that machine dependence can contribute to the total uncertainty that accompanies model intercomparison analyses by an amount that can be at least equal to the one presented
in this study.

*Code availability.* Access to the model code used in the manuscript has been granted to the editor. The source code of the UM model is available under licence. To apply for a licence go to http://www.metoffice.gov.uk/research/modelling-systems/unified-model. JULES is available under licence free of charge, see https://jules-lsm.github.io/. The NEMO model code is available from http://www.nemo-ocean.eu. The model code for CICE can be downloaded from https://code.metoffice.gov.uk/trac/cice/browser.

*Data availability.* Access to the data used in the manuscript has been granted to the editor. The CMIP6 PI simulation run by the UK Met
Office will be made available on the Earth System Grid Federation (ESGF)

(https://cera-www.dkrz.de/WDCC/ui/cerasearch/cmip6?input=CMIP6.CMIP.MOHC.UKESM1-0-LL), the data repository for all CMIP6 output. CMIP6 outputs are expected to be public by 2020. Dataset used for the analysis of the PI simulation ported to ARCHER can be shared, under request, via the CEDA platform (https://help.ceda.ac.uk). Please contact the authors.

*Author contributions.* M.V.G ran the ARCHER simulation, processed the data and carried out the scientific analysis with the contribution
of L.C.S and D.S. M.V.G carried out tests with simple model described in section 2. G.L and R.H ported the PI simulation to the ARCHER supercomputer, provided technical support and advised on the nature of machine-dependent processes. All authors revised the manuscript.

*Acknowledgements.* M.V.G. and L.S. acknowledge the financial support of the NERC research grants NE/P013279/1 and NE/P009271/1. This work used the ARCHER UK National Supercomputing Service (http://www.archer.ac.uk). Authors acknowledge use of the UK Met
Office supercomputing facility in providing data for model comparisons.



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





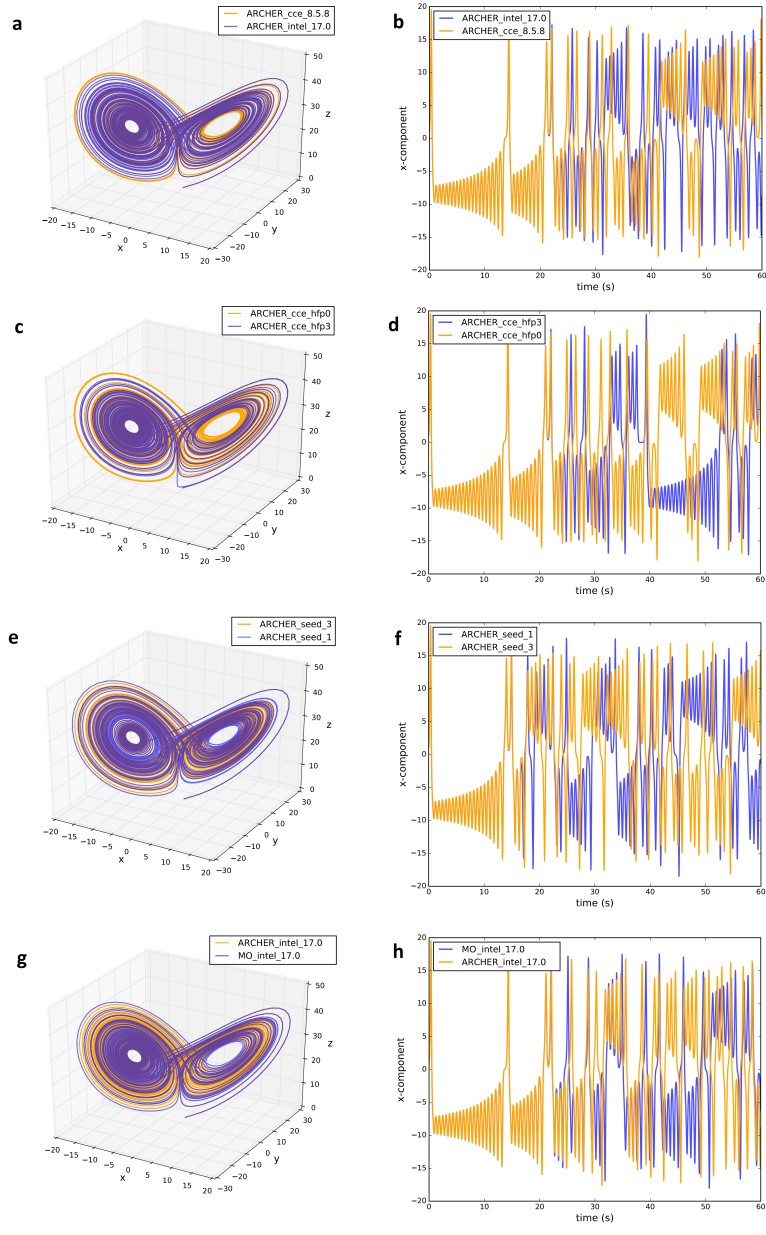

**Figure 1.** Attractor (left-hand side) and time-series of the x-component (right-hand side) of the 3D Lorenz model for simulations run on ARCHER using: the cce8.3.4 and intel17.0 compilers (a, b), same compiler but different level of floating-point optimization (c, d), same compiler and compiling options but different seed for random number generator (e, f). g and h are the Lorenz attractor and the x-component time-series for the Lorenz model run on MO and ARCHER using same compiler and compiling options.



**Figure 2.** 200-year means and corresponding SNR of ($PI_{MO}$ - $PI_{AR}$) differences for NH SST (a, b), SH SST (c, d), NH SIC (e, f) and SH SIC (g, h).





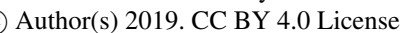

**Figure 3.** 200-year means and corresponding SNR of ($PI_{MO}$ - $PI_{AR}$) differences for SAT (a, b), SW TOA (c, d), LW TOA (e, f) and P (g, h).





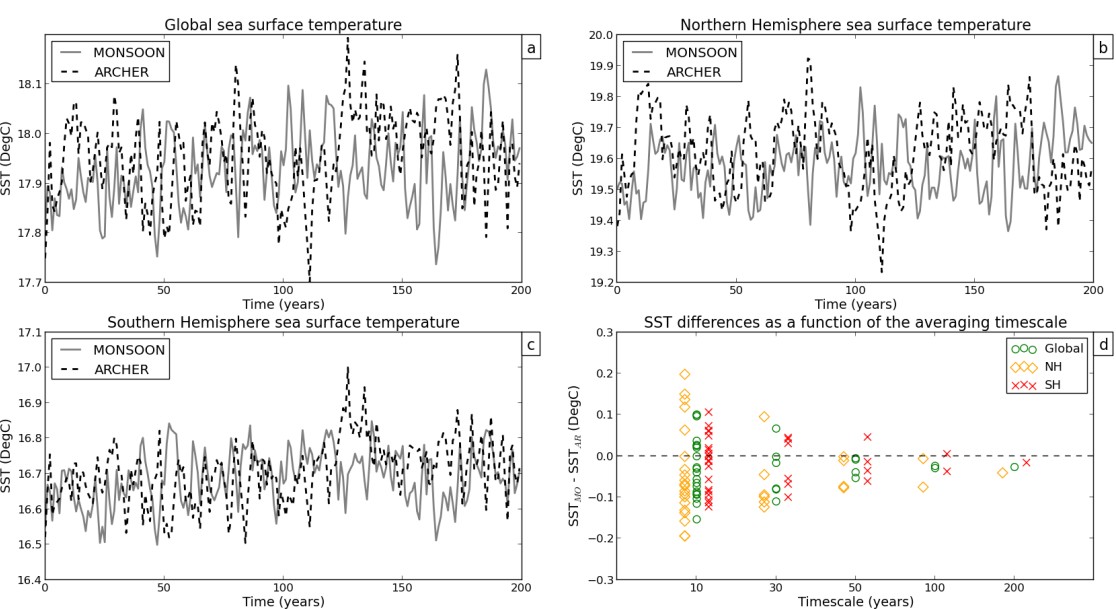

**Figure 4.** Annual-mean time-series of Global SST (a), Northern Hemisphere SST (b) and Southern Hemisphere SST (c) for PI$_{MO}$ (grey line) and PI$_{AR}$ (dashed line). d shows how SST differences vary as a function of the timescale.





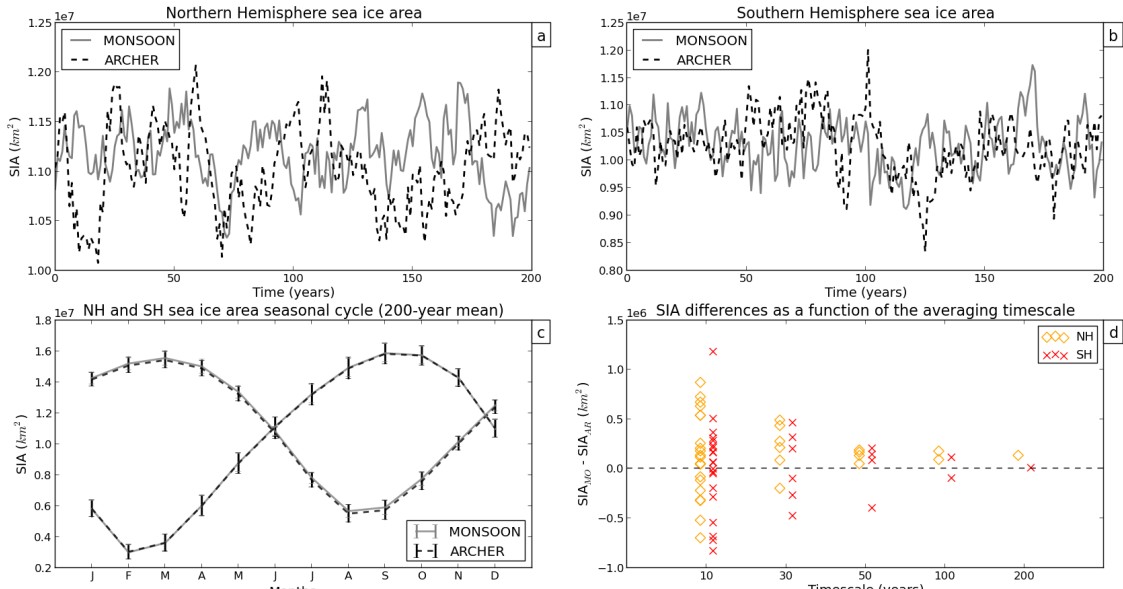

**Figure 5.** Annual-mean time-series of Northern Hemisphere SIA (a) and Southern Hemisphere SIA (b) for $PI_{MO}$ (grey line) and $PI_{AR}$ (dashed line). The 200-year mean of the NH and SH SIA seasonal cycle is shown in c. d shows how SIA differences vary as a function of the timescale.





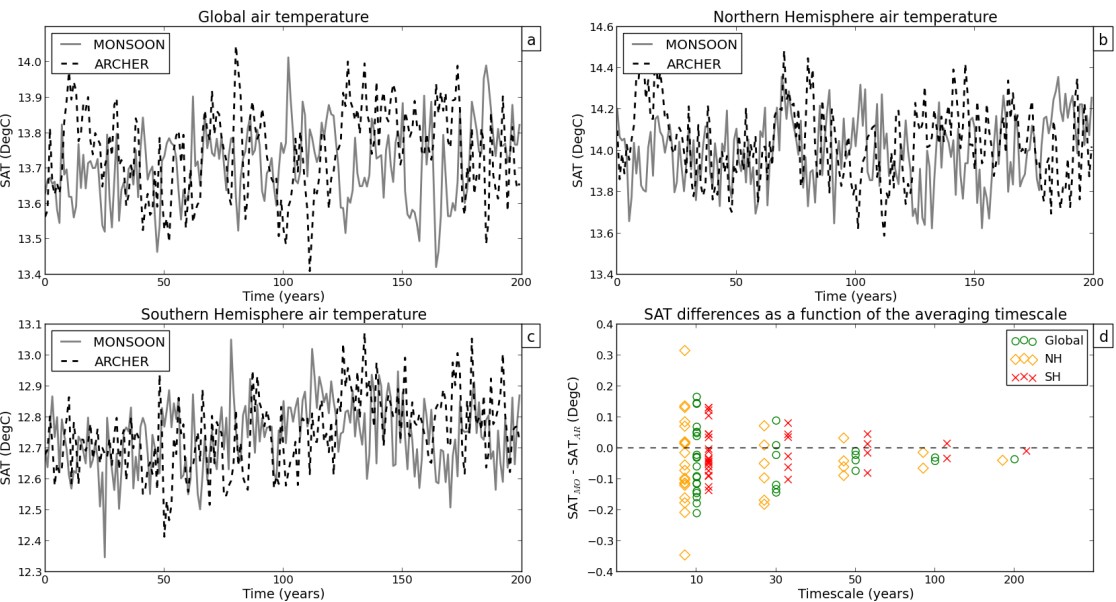

**Figure 6.** As in 4 but for SAT.





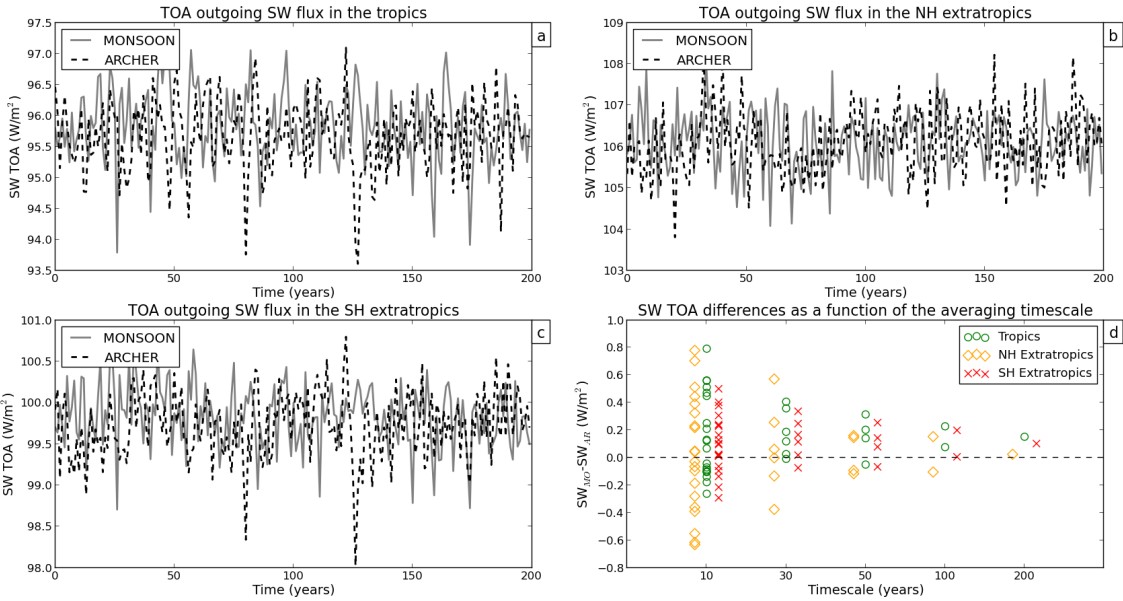

**Figure 7.** Annual-mean time-series of SW TOA in the tropics (a), SW TOA in the Northern Extratropics (b) and SW TOA in the Southern Extratropics (c) for $PI_{MO}$ (grey line) and $PI_{AR}$ (dashed line). d shows how SW TOA differences vary as a function of the timescale.





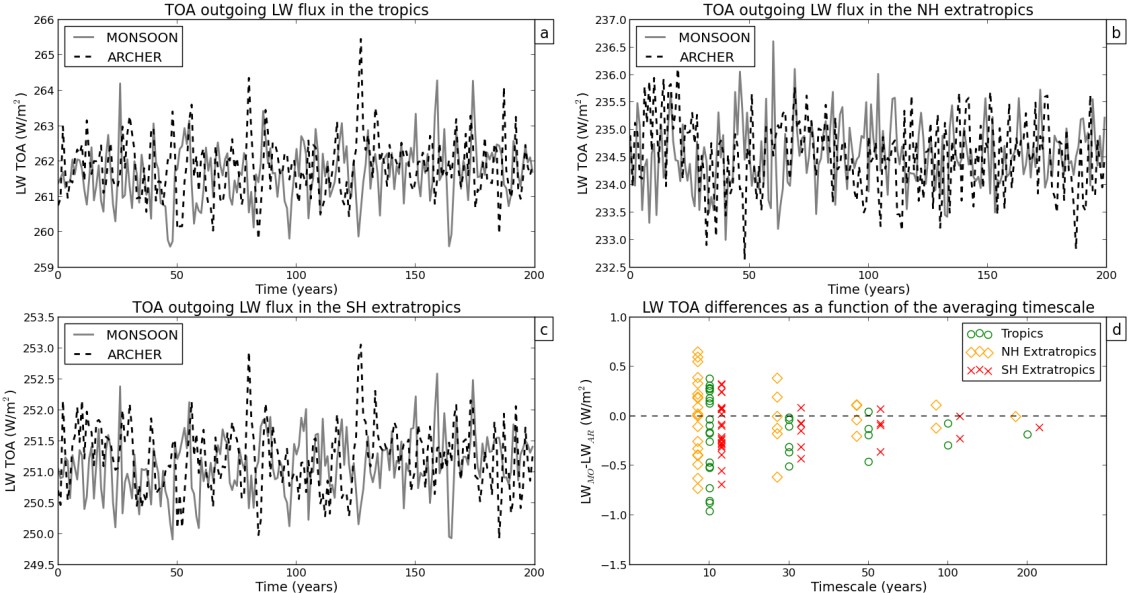

**Figure 8.** As in 4 but for LW TOA.

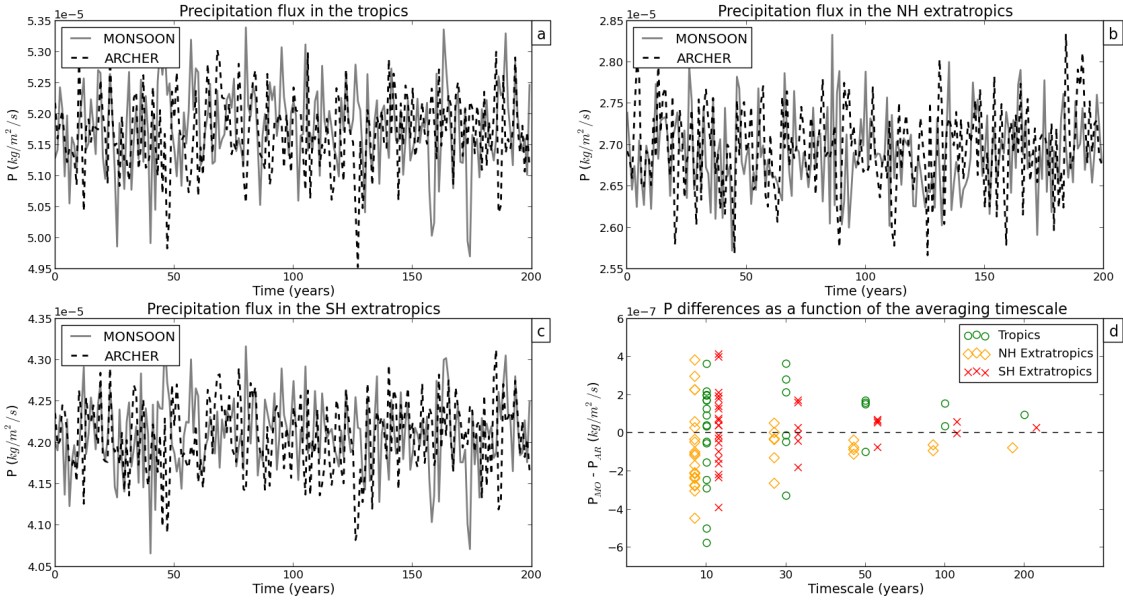

**Figure 9.** As in 4 but for P.



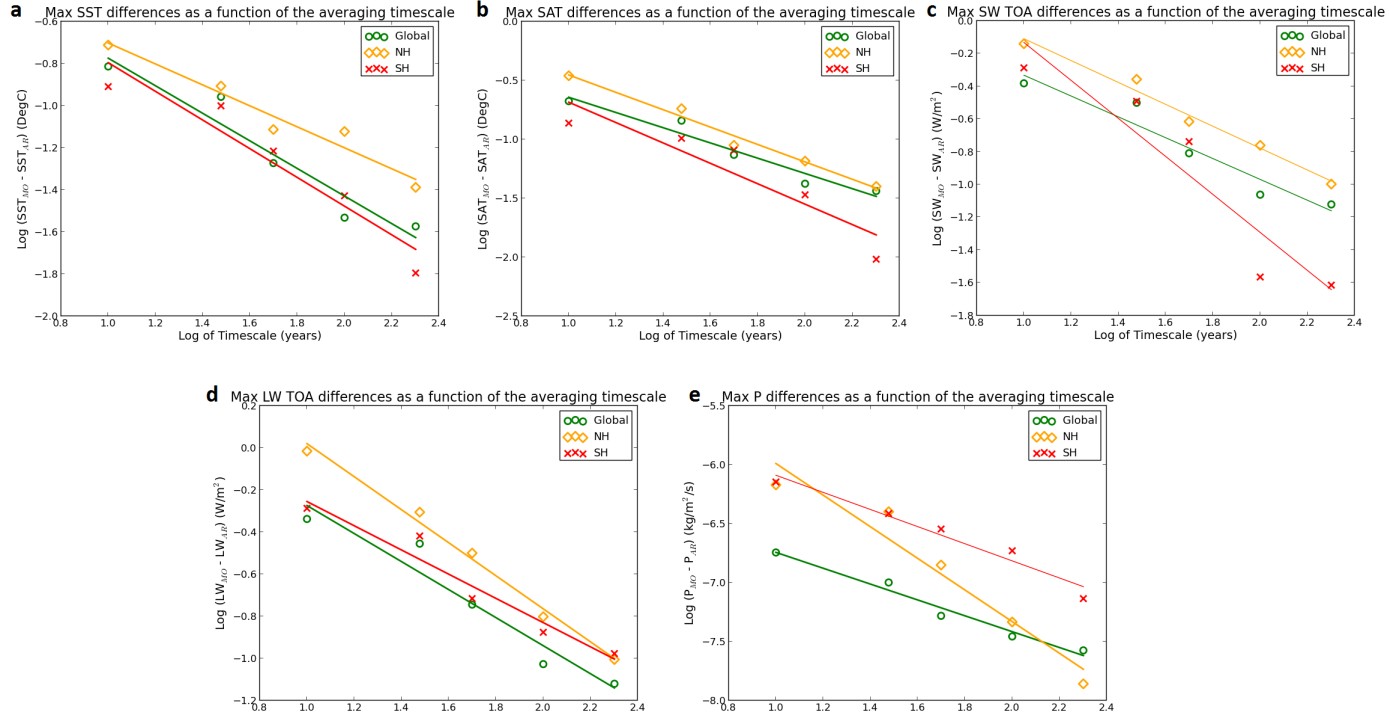

**Figure 10.** Log-log plots of SST (a), SAT (b), SW TOA (c), LW TOA (d) and P (e) representing maximum ($\text{PI}_{MO}$ - $\text{PI}_{AR}$) differences as a function of the timescale. All the variables were averaged globally (green circles) and over the SH (red crosses) and NH (yellow diamonds) Hemispheres. The straight-lines represent the best fit lines for the data obtained by linear regression.





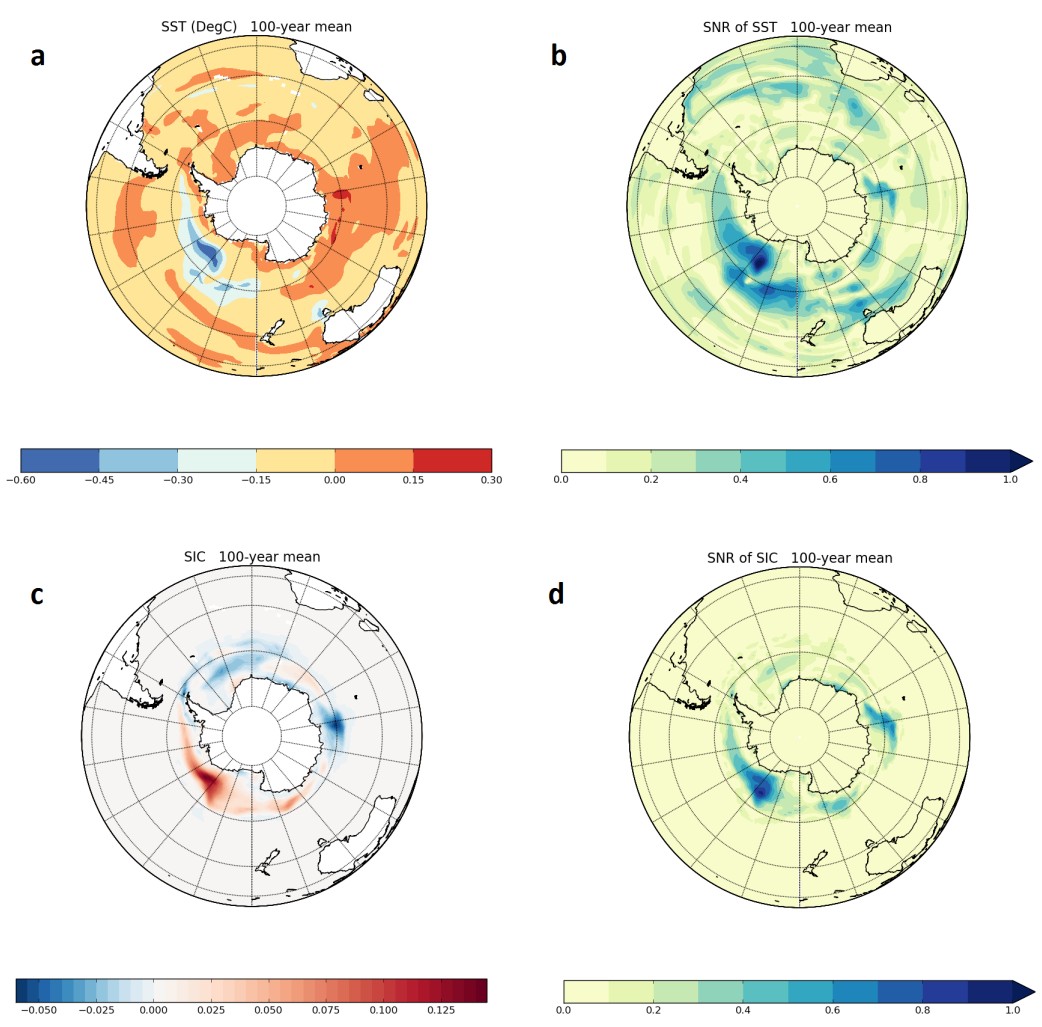

**Figure 11.** 100-year means and corresponding SNR of ($PI_{MO}$ - $PI_{AR}$) differences for SH SST (a, b) and SH SIC (c, d).





**Figure 12.** The ENSO (a), winter NAO (b) and annual SAM (c) indexes for $PI_{MO}$ and $PI_{AR}$. In a, a 3-month running mean was applied to the ENSO signal and values greater/smaller than or equal to $\pm\,0.5$ are shaded in orange/green.