# Peer review of "Machine dependence and reproducibility for coupled climate simulations: The HadGEM3-GC3.1 CMIP Preindustrial simulation"

_Geoscientific Model Development, 2019_

## Referee Comment (RC1) · Anonymous Referee #1 · 12 Jun 2019

The article by Maria-Vittoria Guarino, Louise C. Sime, David Schroeder, Grenville M. S. Lister and Rosalyn Hatcher is about estimating the potential changes in the climate simulated by a climate model when run on different computers. The subject is particularly interesting and as mentioned by the authors, probably overlooked.

However, I fundamentally disagree with their approach and find their demonstration and arguments confusing, leading to wrong conclusions. I do not recommend the publication of the article in GMD. You will find below some general comments followed by specific comments explaining why.

General comments

[Figure]

The authors assume that computing the difference between two preindustrial simulations run from the same initial conditions and with the same boundary conditions is a measure of machine dependence (last paragraph of the introduction, page 3, lines 4-5; page 9, lines 14-15; page 10, lines 32-34; page 11, lines 10-11). This is fundamentally wrong: it is primarily a measure of differences due to internal variability. The goal of this study is to prove that the difference between PIMO and PIAR cannot be due to internal variability. In other words, the null hypothesis of the test is : Âń the differences between PIMO and PIAR are due to internal variability Âż. Rejecting this null hypothesis can be a proof of machine dependence.

As illustrated in section 2 (an excellent illustration of the influence of two different computing environments on the trajectory in a Lorenz attractor, that should be put to the credit of the authors), the same chaotic model on two different machines will not follow the same trajectory after a de-correlation time because the differences in the way the operations are treated, the differences in roundings (among others) will act as these were infinitesimal perturbations. The model trajectories will thus diverge, and this is expected. But we have no mean to say whether this is due to the computer, or to the behavior of the chaotic equations that would behave the same way on one single computer with added random perturbations.

One way to show that the model is different when run on two different machines is to compare the statistical properties of both attractors, for instance mean and variance, and demonstrate that the differences are not due to random processes (internal variability) or a lack of sampling. This last point is addressed in the paper but with the wrong approach. The sentence "We will focus on estimating how long constant-forcing climate simulations should be for machine dependence uncertainty to become negligible" (page 3, lines 4-5) illustrates that the authors didn't understand what estimating machine dependence is about. Increasing the length of the time series to estimate the mean and variance will allow stabilizing these estimated statistics (i.e. reducing the uncertainty on the estimates). The presence of low-frequency components (mainly

the ocean), depending on the climate model considered (they don't all have the same internal variability) implies using multi-centuries time series, which is what the authors do; but the only thing that should decrease with the increasing length of the time series is the uncertainty on the estimates of the statistical properties of the climate of the model. The machine dependence should actually become more and more significant (come out of the noise due to internal variability) with the increasing length of the time series considered, if it truly had an influence. As well, the sentence page 11 lines 10-11 illustrates the confusion between internal variability and machine dependence.

The whole section on the physical implications is out of subject. You are speculating on differences due to internal variability.

Eventually, the authors make suggestions about CMIP6 that are well beyond the reach of their results (page 11, lines 18-23). The analyses of the paper only illustrate that internal variability can cause differences between two climatologies. CMIP6 encourages the modeling groups to provide ensembles with as many members as possible to study internal variability and intercompare the models, and we already see modeling groups providing tens of members of in the CMIP6 DECK historical experiment.

Machine dependence is an important subject in climate modeling and I agree that we should keep an eye on it routinely. However, from a CMIP multi-model comparison point of view, these potential differences are much smaller than the inter-model differences: to my knowledge, a true machine dependence with a CMIP climate model is still to be proven (potentially because they are very small). I invite you to read the publication by Milroy et al (2018) (https://www.geosci-model-dev.net/11/697/2018/) to see an interesting approach to compare model results on different computers. If you read French (I assume that the second co-author does, apologies if you don't), you can also have a look at this technical note from the Laboratoire de Météorologie Dynamique: http://cmc.ipsl.fr/images/publications/technical_notes/jerome_LMDZinfo9.pdf

Specific comments

Page 2, lines 25-27: machine dependance implies that the climate of a given model run on computer A is different from the climate of the model run on computer B. Therefore, machine dependance is a source of uncertainty for the climate models (if we can show that it is significant). Starting from here, reducing this uncertainty would imply showing that the climate simulated on computer A is more relevant than the climate simulated on computer B. Answering this question totally depends on the scientific question you ask. Selecting climate models, or weighting the projections according to selected criteria (like the so-called emergent constraints) is a research subject in itself, and until now the different attempts have not been able to drastically reduce the model uncertainty on future projections. Additionally, the differences between the climate simulated by the CMIP(5,6) models are easily shown by intercomparison results (see Chapter 9 IPCC-AR5) when it is pretty tricky to evidence a machine dependence (potentially much smaller than differences between the models). Therefore, I agree that the machine dependence uncertainty could be estimated someway and taken into account, but I think it is wrong to say that it could be removed by running all the models on the same machine. If we can't select a model today on climate-based criteria and comparisons with observations, I don't see any chance to select one computer. Running all the models on the same computer is not reducing or removing machine dependence: it's ignoring it (what we do today), which is fundamentally different.

Page 6, lines 13-20: I got really confused by this paragraph. You say Âń [. . .] using a chronological order in the strictest sense is meaningless because every 10 years segment is equally representative of the pre-industrial climate variability Âż. And the last two sentences of the paragraph (lines 19-20) contradict this statement. Starting from the same initial conditions, PIMO and PIAR take different trajectories after a couple of time steps, when the chaotic nature of the model takes over. The atmosphere is the first component to diverge (less than one day, much less than the 1 to 2 years mentioned page 8 line 16), and then the slow components of the climate model (namely the ocean and the soil) will keep similar trajectories for as long as their correlation time (depending notably on the geographical region considered) and then will diverge. Studies

of the potential decadal predictability of the climate system in a so-called perfect model framework have shown that this correlation time is around a decade (model dependent). After around one decade, the memory of the initial conditions will be lost. There is thus no justification to compare the same dates. Therefore, in absence of more precise explanation, it is not Âń helpful in investigating the impact of machine dependance on model results Âż. I would add that this kind of vague formulation doesn't have it's place in a scientific paper.

Section 3.2: the SNR measure you use to estimate if the mean of PIMO is different from the mean of PIAR should come with a test to determine more objectively when the difference becomes significant. For instance, if you compute your SNR on n years, you could sample 100 couples of n random years (bootstrap) in the same simulation to check the distribution of the SNR when computed between two (random) periods of the same simulation. This would give you an estimate of the influence of internal variability on your SNR (but would also need a longer simulation for this). You can then compare the SNR computed between PIMO and PIAR with the distribution of the SNR within the same simulation and estimate the probability to have the same value between two periods of the same simulation. An alternative would be the use of the Student t-test on the difference of mean, and the Fisher F-test on the difference of variance. The paragraph page 7 lines 8-14 is totally confusing for me. Extract only the information that brings concrete elements to the debate, and remove the rest. Last point: I don't understand where the sqrt(2) comes from in the SNR formula. . . I would need a reference or an explanation for that.

Section 4.1: the presentation of your results is an illustration of the use of the SNR without knowing when it actually shows a true difference: you say that there are values of the SNR close to 1. Fair enough, as long as those differences are lower than one we should not care about them because they do not show any significant difference (according to your definition of the SNR). And then you talk about SNR values of 0.9, 0.7, that are supposed to be close to 1. How close from one are those results? From

which threshold are we supposed to take these results as proof of a true difference due to machine dependence? Following your definition of the SNR, you should not even mention it and just conclude that the difference in mean between PIMO and PIAR for the concerned variables is not significant. That's it. And not care about the physical explanations of this result.

Page 8, line 30 : you conclude much more than your results show ! Your results show that there is no difference only for the diagnostics that you've done, and those diagnostics have limited implications. You can only conclude that Âń according to our analyses of the SNR, the mean climate for the variables assessed here is not different between PIMO and PIAR. The porting did not affect the climatology of the models for those variables."

Page 8 line 31 to page 9 line 4: I'm afraid you came to this conclusion because you don't understand the problem, or try to sell your results way beyond what they show. Same for last paragraph of section 4.2 : you may just have a SNR larger than one (how larger? Is it significant?) because of the low-frequency internal variability in your model. This means that you need longer simulations to properly assess this question.

---

## Referee Comment (RC2) · Anonymous Referee #2 · 13 Jun 2019

This paper examines the difference in two control simulations with a state-of-the-art climate model using nominally the same code but on different machines (with different compilers, chips and libraries). The rate at which the simulations diverge from each other with the same initial coniditions is consistent with the expected divergence associated with a sensitive dependence to initial conditions, but over the longer term, there are no detected differences in the climatology.

Historically, it has been the case that some climate model simulations have had a climatolgogy that varied as a function of machine platform, though this has rarely been discussed in the literature (though see a very recent example here: https://www.geoscimodel-dev-discuss.net/gmd-2019-91/). Generally this is an indication of bugs in the code that manifest themselves differently and systematically so, on a different architecture, and thus would have been problematic had this been discovered here.

However, the results presented here demonstrate that the climatology of the two simulations is the same - given a long enough averaging period the simulations are indistinguishable. This is a good result, however, it is not the conclusion that the authors come to.

Part of the problem I think is conceptual. In an ideal simulation of a chaotic system on two different architectures with identical initial conditions, the differences in the machines will manifest themselves as machine-level rounding differences spread throughout the calculation (as shown in section 2). Since the GCM is a simulation with sensitive dependendce on initital conditions, this will place the simulations on slightly adjacent trajectories which would then be expected to diverge with whatever Lyapunov exponent is relevant. Subsequent time-steps will simply repeat the exercise (i.e. perturbing the initial coniditons for the next time step by machine precision). I do not see how this will produce anything fundamentally different from a standard initial condition ensemble. Therefore the question to be asked of the two simulations discussed here is whether the simulations are distinguishable from an IC ensemble on a single machine, not whether they diverge at all. I note that this is the standard used in Hong et al, 2013 in a slightly different context. I am a little puzzled that the authors are not seeing this (especially given the statement on p11. line 24). Given this, the analysis in sections 3 and 4 are of little interest.

Thus I do not think the current paper is publishable. A re-conceptulaised analysis of these runs and this issue might be acceptable, but that would be quite a different paper.

Minor points:

p2 line 31. The climate modeling community spends an enormous amount of effort to ensure bitwise reproducibility for testing and development purposes. The point made

in the Liu et al 2015 paper is whether more effort should be spent to ensure it in a broader context (i.e. over years and across platforms), not whether it's worth doing at all.

p2 line 35. "the uncertainty attributable to machine-dependent processes" - I disagree. The authors have not attributed this at all.

p8. line 34. No. It suggests only that there are adjustment times longer than 100 years in the climate system.

p11. line 10. The authors are merely assuming that the differences between the two runs are due to 'machine dependence' - this is begging the question.

p11. line 16. There is no 'machine-induced bias' in these runs (bias is a difference in the long-term means).

p11. line 30-34. I am all for greater ensemble sizes (which actually, many groups are doing - i.e. the CESM Large ensemble, or the CCCM large ensemble), but this is again related to standard IC-related dependence, not machine dependency.

p12. line 8. "suspicious"??? This is a very odd term to apply.

---

## Author Comment (AC1) · 7 Aug 2019

We thank the referees for the time they have taken to provide valuable comments on our manuscript, and the editor for giving us the opportunity to address the concerns raised by the referees.

We believe we have made the best use of the reviewers' comments to improve our manuscript and deliver our objective more clearly. We hope the responses (attached) address the referees' comments.

Please also note the supplement to this comment:

https://www.geosci-model-dev-discuss.net/gmd-2019-83/gmd-2019-83-AC1-supplement.pdf

**Supplement:**

We thank the editor and referees for the time they have taken to provide valuable comments on our manuscript, the comments have enabled us to make substantial clarifications and other improvements to the manuscript.

We note that the major concern from both referees was on the role of internal variability on our results. In this, the referees raise a valid point. The nature and the purpose of our work was misunderstood by both referees because it was previously insufficiently clear. We have taken the time to clarify the purpose of the paper by rewriting the introduction and have rewritten several other sections within the manuscript, as was required. It is not our aim to fully separate between machine dependence uncertainty and internal variability. Instead we aim to firstly, explore the influence of machine dependence for our UK CMIP6 model HadGEM3-GC3.1 simulations and secondly, give the wider modelling community general practical guidance when simulations have to be run on different machines – as has frequently been the case for the paleo-modelling MIPs – and is the case for our HadGEM3-GC3.1 simulations.

To make our responses complete, the revised manuscript is provided at the end of the document.

**Responses to Referee n1**

- ## General Comments

1) "The authors assume that computing the difference between two preindustrial simulations run from the same initial conditions and with the same boundary conditions is a measure of machine dependence. This is fundamentally wrong: it is primarily a measure of differences due to internal variability."

Ref.n1 is right in that we call "machine dependence" those differences that we can observe by simply running a same climate simulation (same model, set-up, forcing etc.) on two different machines.

The context in which the term "machine dependence" is used in the paper is explained in section 2, where we explain possible reasons that can cause the solution in the two cases to evolve differently. We conclude section 2, p4 line 1, clearly stating that differences are triggered by machine-dependent processes (compiler, optimization and etc.) *but eventually exist because of the chaotic nature of the system* (internal variability). Thus, the differences among the two runs analysed throughout the paper are not a measure of the machine dependence over the internal variability (the type of machine dependence Ref.1 refers to) but a measure of how the internal variability responds to the different computing environment in the two cases (see also response to Ref.n2 point 1).

The purpose of our work is to find out for how long we should run a simulation (or analyse its results) on the ARCHER HPC platform to capture/sample the same climate variability exhibited by the reference simulation on the Met Office supercomputer (as we know the two runs do not bit-compare).

To clarify, we reformulated section 1, and adopted a change of terminology, requiring a change of the title of the manuscript alongside a revision of the results and conclusions sections.

2) "I invite you to read the publication by Milroy et al (2018) (https://www.geosci-model-dev.net/11/697/2018/) to see an interesting approach to compare model results on different computers."

The porting of the HadGEM3-GC3.1 model from the Met Office computing platform to the ARCHER platform was tested as a part of routine tests performed by the UK Met Office and NCAS-CMS teams. As the purpose of our work is not to assess the porting of the specific PI simulation, the results of the porting and of its testing are not included in the manuscript. However, we added a paragraph (revised manuscript p5, line 5), explaining model porting and testing.

For completeness, here we provide additional details on the procedure used by the UK team: consistency across simulations on different computers are routinely tested by running 50 ensemble members (each 24 hours long) on both platforms. Each ensemble member is created by adding a random bit-level perturbation to a set of variables of the model initial conditions. These variables are: x- and y- components of the wind, air potential temperature, specific humidity, mass fraction of cloud, air pressure, long-wave and sort-wave radiation.

Variables from each set of ensembles are then tested for significance using a Kolmogorov Smirnov test to determine whether they can be assumed to be drawn from the same distribution.

These tests did not reveal any significant problem with the porting of the HadGEM3-GC3.1 model (please note that these tests were not performed directly by the authors, thus we cannot share the results). Note also that this method cannot detect code bugs, which may cause a same model to behave differently on different machines. In this respect, when we conclude at page 8 line 1-3 that the long-term statistics of the two runs are similar, we provide a strong indication that the HadGEM3-GC3.1 model does not suffer from code bugs giving different outcomes depending on the computing environment.  We have now added a sentence highlighting this at page 8, line 18-19:

"Our results also provide a strong indication that HadGEM3-GC3.1 does not suffer from code/compiler bugs that would make the model behave differently on different machines."

3) The whole section on the physical implications is out of subject. You are speculating on differences due to internal variability.

Please see response to Ref.n2 point 4.

- **Specific Comments**

4) "Therefore, I agree that the machine dependence uncertainty could be estimated someway and taken into account, but I think it is wrong to say that it could be removed by running all the models on the same machine. If we can't select a model today on climate-based criteria and comparisons with observations, I don't see any chance to select one computer. Running all the models on the same computer is not reducing or removing machine dependence: it's ignoring it (what we do today), which is fundamentally different.

The Introduction has been substantially changed and the paragraph the reviewer refers to has been removed.  Please refer to revised manuscript.

5) "Page 6, lines 13-20: I got really confused by this paragraph. You say "using a chronological order in the strictest sense is meaningless because every 10 years segment is equally representative of the pre-industrial climate variability". And the last two sentences of the paragraph (lines 19-20) contradict this statement.

We agree with the reviewer that this was written in a confusing way, to address this, Lines 19-20 are removed.

6) "Section 3.2: the SNR measure you use to estimate if the mean of PIMO is different from the mean of PIAR should come with a test to determine more objectively when the difference becomes significant. For instance, if you compute your SNR on n years, you could sample 100 couples of n random years (bootstrap) in the same simulation to check the distribution of the SNR when computed between two (random) periods of the same simulation. This would give you an estimate of the influence of internal variability on your SNR (but would also need a longer simulation for this)... An alternative would be the use of the Student t-test on the difference of mean, and the Fisher F-test on the difference of variance."

Alongside computing SNR, PI_MO − PI_AR differences were tested using a 2-tailed Welch's T-test (where our H0 is mean_MO = mean_AR).

In the paper, only results based on SNR are shown. That is because, while most of the time the T-test and the SNR-based analysis gave us the same answer, in a few instances we had discordant results (see Figure 1). Overall, the behaviour detected in comparing the two methods was:  t-test indicating significant differences despite SNR < 1. Bearing in mind that what a t-test really signifies is: IF H0 were true the probability of obtaining, in repeated experiments, the observed mean_MO − mean_AR value is less than 0.05, we made the decision of keeping on using SNR as, in our view, this quantity has a greater physical weight and can more readily be interpreted.  Additionally, by doing so we chose the most conservative method of the two (i.e. the one that less frequently points to significant differences).

[Figure]

Figure 1 a) SNR computed as in manuscript eq (2) for PI_MO − PI_AR SST differences (on a 50-years period): maximum value for SNR is 0.9, indicating that there are differences but SNR still below the threshold value of 1. b) 2-tailed Welch's T-test on same data: where SNR ~ 0.7 - 0.9, the p-value is less than alpha. According to this, differences should be considered significant.

7) "... as long as those differences are lower than one we should not care about them because they do not show any significant difference (according to your definition of the SNR). And then you talk about SNR values of 0.9, 0.7, that are supposed to be close to 1. How close from one are those results?"

We address this comment by modifying the manuscript so that we analyse results in regions only where SNR ≥ 1. See for example p6 line 24, p9 line6, p9 line 29.

**Responses to Referee n2**

- **General Comments**

1) "Part of the problem I think is conceptual… Subsequent time-steps will simply repeat the exercise (i.e. perturbing the initial conditions for the next time step by machine precision). I do not see how this will produce anything fundamentally different from a standard initial condition ensemble."

A rather detailed explanation of how the machine may affect the numerical solution is given in section 2. However, it is clear that the previous version did not fully successfully communicate which questions our analysis and manuscript address (see below).

Internal variability is usually assessed in climate simulations via two methods: ensemble members for varying-forcing simulations and long centennial runs for constant-forcing simulations. In the first case, the question is "how many" ensemble members are needed to sample correctly the climate variability, in the second case the question is "how long". While the obvious answers are the more the better and the longest the better, one may ask what is the minimum number of ensembles, or the minimum simulation length required, that would guarantee an acceptable result.

Our work finds its reason in this context, i.e. we want to know for how long we should run a simulation (or analyse its results) on the ARCHER HPC platform to capture the same magnitude of climate variability exhibited by the reference simulation on the Met Office supercomputer (see also responses to Ref.n1 point 1), as we know that the two runs do not bit-compare.

We have now made this clear in the manuscript. Page 2, line 8-17:

"In this paper, we investigate the behaviour of the UK CMIP6 Preindustrial (PI) control simulation with the HadGEM3-GC3.1 model on two different High Performance Computing (HPC) platforms. We first study whether the two versions of the PI simulation show significant differences in their long-term statistics. This answers our first question of whether the HadGEM3-GC3.1 model gives different results on different HPC platforms.

Machine-dependent processes can influence the model internal variability by causing it to be sampled differently on the two platforms (i.e. similarly to what happens to ensemble members initiated from different initial conditions). Therefore, our second objective is to quantify discrepancies between the two simulations at different time-scales (from decadal to centennial) in order to identify an averaging period/simulation length for which the two simulations return the same internal variability.

Note that the PI control simulation is a constant-forcing simulation. Therefore, no ensemble members are required for such experiment because, provided that the simulation is long enough, it will return a picture of the natural variability."

and page 8 line 19-24:

"The large differences observed on time-scales shorter than 200 years are a direct consequence of machine-dependent processes (compiler, machine architecture etc., see section 2 and 3.1 for details),

but eventually exist because of the chaotic nature of the system. The two simulations behave similarly to ensemble members initiated from different initial conditions. Therefore, they exhibit different phases of the same internal variability but over longer time-scales differences converge to zero (Figure 4 - 9)."

Please see also responses to Ref.n1 point 2 for details about the model porting and tests performed on IC ensembles, and responses to Ref.n1 point 6 for details on differences on single machine.

2) "Therefore the question to be asked of the two simulations discussed here is whether the simulations are distinguishable from an IC ensemble on a single machine, not whether they diverge at all".

We added a section to the manuscript; see page 9, starting at line 15 (and Figure 11e and 11f). In this section, we now show the signal-to-noise ratio computed by taking the differences between two 100-year periods for each simulation (note that each 100-year period can be considered as an ensemble member run on the same machine). This shows that differences between the two machines are comparable to differences among ensemble members run on a single machine. This confirms that the differences we observe between PI_MO and PI_AR, although triggered by the different computing environment, are largely dominated by the internal variability. This also highlights that 100 years is a too-short length for constant-forcing simulations on the same, or on a different, machine.

3) "However, the results presented here demonstrate that the climatology of the two simulations is the same - given a long enough averaging period the simulations are indistinguishable. This is a good result, however, it is not the conclusion that the authors come to."

The first result presented in the paper (in the revised manuscript page 7, line 5-7) shows that on multi-centennial time-scales the differences are not significant and the long-term statistics of the two runs are similar. We have now given more strength to this result. Page 8, line 15-19:

"In summary, although large differences can be observed at smaller time-scales (see next section for further discussion), the climate of PIMO and PIAR is indistinguishable on the 200-year time-scale. We thus conclude that simulations using the HadGEM3-GC3.1 model are reproducible on different HPC platforms, provided that a long-enough simulation length is used. Our results also provide a strong indication that HadGEM3-GC3.1 does not suffer from code/compiler bugs that would make the model behave differently on different machines "

4) "Given this, the analysis in sections 3 and 4 are of little interest."

Section 4 answers our question of how long we should run a copy–simulation on the ARCHER platform. In this section, we show that only when using a 200-year averaging period we capture the same internal variability in both simulations. This is the main result of our paper, which implies not only that running a constant-forcing simulation for less than 100 years may potentially lead to different outcomes but also that running it for longer may not be necessary (this applies however only to those variables considered in the paper).

The additional analysis done on the ENSO signal is meant to be a demonstration of what physical process can cause such big differences. However, we agree with the reviewers that it is not surprising

that a low-frequency process like ENSO is the one still showing differences on a 100-year timescale. This remark was added at page 9, line 13-14.

As both reviewers agree that this section is of lesser interest, we have shortened section 4.2 in the revised manuscript.

Finally, we believe that the reference to the CMIP project is appropriate. CMIP is much more than Historical and Scenario simulations (for which ensembles are requested). Many individual MIPs run constant-forcing simulations with varying lengths depending on time availability, computational resources and length requirements. The CMIP6 minimum run length requirement for many of the Model Intercomparison Projects (MIPs) is 100 years. Our results suggest that 100 years may not be long to capture correctly the internal variability of the HadGEM3-GC3.1 model.

- **Specific Comments**

Please refer to the revised manuscript, as Introduction and Conclusions have substantially changed since the previous version.

[revised manuscript text omitted]

---

## Referee Report (RR1)

**Review of the submitted article « Machine dependance and reproducibility for coupled climate simulations: The HadGEM3-GC3.1 CMIP Preindustrial simulation »**

The article by Maria-Vittoria Guarino, Louise C. Sime, David Schroeder, Grenville M. S. Lister and Rosalyn Hatcher is about estimating the potential changes in the climate simulated by a climate model when run on different computers, and the influence of internal variability in this type of study.

Overall, the paper has largely improved compared with the first version. However, I still find the same tendency along the manuscript for over-interpretation of the results, too many suggestions and not enough clear use and interpretation of the actual results, not enough precision and details on the methodology, and even worse, a wrong initial statement as the starting point of section 4.2 (minimum length for a piControl DECK simulation in CMIP6 of is 100 years when it is actually 500 years), that is the base of one of the main conclusions of the paper.

Unfortunately, following those elements, I still not recommend the manuscript for publication in GMD.

You will find below some specific comments supporting my decision.

Page 4, lines 9-10: 'all' the CMIP6 experiments are not analyzed against the piControl. Replace 'all' with 'many'.

Page 5, lines 11-13: the sentence "However this method cannot detect code bugs, which may cause a model to behave differently on different machines" is incorrect. First, the method you present is suited to check the resolution of the equations of the model on a period shorter than one day. It could thus detect what you call 'bugs'. And second: what are you actually calling bugs? In your study you are trying to detect whether the resolution of the equations of the model on two different machines could end up with two different climates. But which one is the right one? In case you find that the simulations differ in some way, is there a way to say that one is correct and the other is not? I suggest to replace this sentence with: "However this method is restricted to time scales shorter than one day. The centennial simulations presented in this paper will help understanding whether or not differences can arise on longer time scales in the HadGEM3-GC3.1 model."

Page 6, line 8: I still don't get why you divide your SNR by sqrt(2). I'm not saying that it's not relevant, I just don't find any explicit reason, supported by a reference, or a demonstration, to explain this choice in your manuscript. I ask for clarification in the text (lines 14-15 are not enough, need a reference), especially because you say that MO-AR differences are outside the internal variability range for values greater than one (and not 0.9, or 1.1).

Page 6, lines 9-12: please reconsider this paragraph with this proposition: "When SNR<1, MO-AR differences can be interpreted as fluctuations within the estimated range of internal variability. When SNR>1, the MO-AR differences in mean are outside the expected range of internal variability. It means that we either evidenced a true difference in mean, or that the estimated range of variability is underestimated".

Page 7, lines 13-19: the simple and direct way to explain the behavior of your results is:
- on decadal time scale, the period is too short to adequately sample the longer time scales of the interannual variability; therefore the estimated mean is not stable, and the estimated standard deviation of the simulation is likely underestimated compared with the true standard deviation of the internal variability of the model; it is thus not surprising to have values higher than one when analyzing decadal periods
- on longer time scales, the estimate of the mean and standard deviation converge toward their 'true' values. Accordingly, we see that the differences between and MO and AR become smaller.
- For the 200-year long period, we find no value greater than one. Following this diagnostic, and for the variables we assessed, the results show that there is no significant difference in mean simulated with HadGEM3-GC3.1 on MO and AR

And this is valid only for the mean, and for the variables considered. You can thus reconsider your last sentence (line 18-19) by saying that "Our results show that there is no difference in mean when considering a 200-year long period between AR and MO". Your suggestion is that "the overall physical behavior of the model has not been affected by the porting" is premature regarding your analyses.

Page 8, lines 16-17: I reject your conclusion that "simulations using the HadGEM3-GC3.1 model are reproducible […] long-enough simulation in length is used". You can only conclude from your analyses that the mean of the variables assessed is not different between MO and AR for your piControl simulations. Actually showing that the model is reproducible would require that your diagnostics provide an exhaustive description of the model physical behavior, not only mean, but also variability, teleconnection patterns, trends... You can say that your analyses do not show that the model is not reproducible, and that's already a valuable information, that has not been provided by all the modeling centers (and you should receive credit for this).

Page 8, line 22: I propose "The large differences observed on time-scales shorter than 200-years are a direct consequence of the (potentially underestimated) internal variability of the model, triggered (at least initially) by the machine-dependent processes (compiler […] 3.1 for details)."

Page 8, line 27: this is a major point: because the analyses presented in your manuscript concern piControl runs performed with a fully-coupled GCM (typically one of the DECK experiments, see https://www.geosci-model-dev.net/9/1937/2016/gmd-9-1937-2016.pdf), I assume that your are talking about the minimum length of the piControl run in the DECK (if not, then you really need to add more details to be more specific). The actual minimum length for a piControl run in the CMIP6 DECK is 500 years. Therefore your section starts with a wrong statement, which is a pretty serious mistake.
Indeed, your results suggest that 100 years may not be enough to fully sample the internal variability of HadGEM3-GC3.1. The good news regarding this statement is that the CMIP6 protocol asked for 500 years. The bad news is that several CMIP6 models have much more internal variability than their previous CMIP5 versions, and that 500 years might not be enough. But this is another story.
Also, your conclusions and suggestions have a priori no reason to be applicable to other experiments/MIPs, such as AMIP, historical runs, scenarios, etc… if you want to make on any other given experiment/MIP, be more specific.

Page 8, lines 4-14: I don't understand where you want to go with the 2/3 power law, although the result is surprisingly consistent among the variables. And the "plateau" you describe is supported by three consecutive points on your plots on figure 10, the last one being slightly higher than expected by the line. I would agree that there is a plateau if it was described by more than one single point being higher than expected. And once more, you conclude that this results "suggests" something. I would recommend using your results to "show" things, and stick to what they actually show.

---

## Author Response (AR2)

Please find below our responses to reviewer n1 and n2. We have done our best to address the reviewer comments and clarify any outstanding issue.

**Responses to Referee n2**

This paper is much improved and is now acceptable. Two minor comments below.

We are pleased that the referee is satisfied with our improved version and would like to thank the referee for the helpful comments improving our manuscript.

- **Specific Comments**

   1) "p8. line 27-28. Which MIPs might be affected by this?"

We have added examples of MIPs recommending 100 years or less as minimum run length for their experiments.

For instance, the CMIP6 minimum run-length requirement for a few of the Model Intercomparison Projects (MIPs), excluding DECK and Historical simulations, is 100 years or less and ensembles are not always requested (e.g., some of the Tier 1/2/3 experiments in PMIP (Otto-Bleisner et al., 2017), nonlinMIP (Good et al., 2016), GeoMIP (Kravitz et al., 2015), High-ResMIP (Haarsma et al., 2016), FAFMIP (Gregory et al., 2016) ).

   2) "p9. line 14. Wittenberg et al (2009. 10.1029/2009GL038710) suggests even longer would be necessary."

The reference was added at page 8, line 14:

As ENSO provides a medium-frequency modulation of the climate system, it is not surprising that it takes longer than 100 years for its variability to be fully represented (see e.g., Wittenberg et al., 2009).

**Responses to Referee n1**

The article by Maria-Vittoria Guarino, Louise C. Sime, David Schroeder, Grenville M. S. Lister and Rosalyn Hatcher is about estimating the potential changes in the climate simulated by a climate model when run on different computers, and the influence of internal variability in this type of study.

Overall, the paper has largely improved compared with the first version. However, I still find the same tendency along the manuscript for over-interpretation of the results, too many suggestions and not enough clear use and interpretation of the actual results, not enough precision and details on the methodology, and even worse, a wrong initial statement as the starting point of section 4.2 (minimum length for a piControl DECK simulation in CMIP6 of is 100 years when it is actually 500 years), that is the base of one of the main conclusions of the paper.  Unfortunately, following those elements, I still not recommend the manuscript for publication in GMD.

We thank the referee for acknowledging the improvements and for his/her censorious attitude towards our manuscript, which is really helpful in eliminating some remaining misunderstandings. However, the referee is mistaken in stating that our initial statement about minimum length is wrong. Please find below our response to the specific comments. We added some details to methodology as suggested and made a few minor changes to the manuscript to avoid misunderstandings.

- **Specific Comments**

  1) "Page 4, lines 9-10: 'all' the CMIP6 experiments are not analysed against the piControl. Replace 'all' with 'many'. "

  Changed as requested.

  2) Page 5, lines 11-13: the sentence "However this method cannot detect code bugs, which may cause a model to behave differently on different machines" is incorrect. First, the method you present is suited to check the resolution of the equations of the model on a period shorter than one day. It could thus detect what you call 'bugs'. And second: what are you actually calling bugs? In your study you are trying to detect whether the resolution of the equations of the model on two different machines could end up with two different climates. But which one is the right one? In case you find that the simulations differ in some way, is there a way to say that one is correct and the other is not? I suggest to replace this sentence with: "However this method is restricted to time scales shorter than one day. The centennial simulations presented in this paper will help understanding whether or not differences can arise on longer time scales in the HadGEM3-GC3.1 model."

  The bugs we intended are compiler bugs, i.e. different compilers can interpret a same line of code differently. The method described in the paper, used to test the porting of the HadGEM3 model, is targeted to identify errors such as round-off, computation order, IEEE arithmetic and basic library function errors (and etc.) that would cause the solution to diverge in the two cases immediately. However, errors resulting from, for example, divisions by zeros (which might or might not occur in the first 24 hours of simulations, and could be interpreted differently depending on the compiler), or random seed initialization in parameterization schemes might occur later on in the simulation.
  We accept the referee's suggestion and have replaced the sentence at page 5, line 11-13, as we do not think that additional clarifications on 'compiler bugs' would benefit the paper's scientific discussion.

  However this method is restricted to time scales shorter than one day. The centennial simulations presented in this paper will help understanding whether or not differences can arise on longer time scales in the HadGEM3-GC3.1 model.

  3) Page 6, line 8: I still don't get why you divide your SNR by sqrt(2). I'm not saying that it's not relevant, I just don't find any explicit reason, supported by a reference, or a demonstration, to explain this choice in your manuscript. I ask for clarification in the text (lines 14-15 are not enough, need a reference), especially because you say that MO-AR differences are outside the internal variability range for values greater than one (and not 0.9, or 1.1).

The assumption used in the paper is based on one of the basic properties of variance (i.e. being a not linear operator), the mathematical demonstration is given at the end of this document for readability (please see below). We now mention this at page 6, line 8. The sentence has also been modified for improved clarity and a reference, as requested, has been added:

The signal is represented by the mean of the differences between PI_MO and PI_AR ($\mu_{MO-AR}$) and the noise is represented by the standard deviation of PI_MO ($\sigma_{MO}$), our "reference" simulation. Because of the basic properties of variance, for which $Var_{X-Y} = Var_X + Var_Y - 2Cov(X,Y)$ (Loeve, 1977), we can more conveniently express the noise as $\sigma_{MO} = \sigma_{MO-AR}/\sqrt{2}$ , under the assumptions that PI_MO and PI_AR are uncorrelated ($Cov(MO,AR) = 0$ ), and have same variance ($Var_{MO} = Var_{AR}$). This allowed us to compute SNR on one same grid, and avoid divisions by (nearly) zero when the sea ice field between PI_MO and PI_AR evolved differently, resulting in unrealistically high SNR values along the sea ice edges.
Finally, SNR is defined as:

$$SNR = \frac{\mu_{MO-AR}}{\sigma_{MO}} = \frac{\mu_{MO-AR}}{\sigma_{MO-AR}/\sqrt{2}}$$

A signal-to-noise ratio larger than 1 is commonly associated to the existence of a physical process. This is because such condition certainly implies that the signal (the mean) is larger than the noise (the standard deviation). However, we agree that using a net cut-off value of 1 can be, in itself, unphysical. In fact, values very close to 1 might still indicate an 'emerging' signal.
A short discussion of why also values close to 0.9 might be important was present in the first submitted version of the manuscript. However, at referee's recommendation, we modified the discussion so to focus only on values larger than 1.  Please see also our previous "Responses to referees" document, responses to Referee n1, specific comments 6 and 7, about this.

4)  Page 6, lines 9-12: please reconsider this paragraph with this proposition: "When SNR<1, MO-AR differences can be interpreted as fluctuations within the estimated range of internal variability. When SNR>1, the MO-AR differences in mean are outside the expected range of internal variability. It means that we either evidenced a true difference in mean, or that the estimated range of variability is underestimated".

The sentence at page 6, line 9-12, was modified following the reviewer's suggestion:

When SNR < 1, (PI_MO - PI_AR) differences can be interpreted as fluctuations within the estimated range of internal variability. When SNR > 1, (PI_MO - PI_AR) differences in the mean are outside the expected range of internal variability. This eventuality indicates either a true difference in the mean, or that the expected range of variability is underestimated.

5) Page 7, lines 13-19: the simple and direct way to explain the behaviour of your results is:
- on decadal time scale, the period is too short to adequately sample the longer time scales of the interannual variability; therefore the estimated mean is not stable, and the estimated standard deviation of the simulation is likely underestimated compared with the true standard deviation of the internal variability of the model; it is thus not surprising to have values higher than one when analysing decadal periods

- on longer time scales, the estimate of the mean and standard deviation converge toward their 'true' values. Accordingly, we see that the differences between and MO and AR become smaller.
- For the 200-year long period, we find no value greater than one. Following this diagnostic, and for the variables we assessed, the results show that there is no significant difference in mean simulated with HadGEM3-GC3.1 on MO and AR.
And this is valid only for the mean, and for the variables considered. You can thus reconsider your last sentence (line 18-19) by saying that "Our results show that there is no difference in mean when considering a 200-year long period between AR and MO". Your suggestion is that "the overall physical behaviour of the model has not been affected by the porting" is premature regarding your analyses.

We accept the suggestions and paragraphs at page 7 have been modified as follows:

On decadal timescales, the averaging period is too short to adequately sample the model interannual variability; therefore the estimated mean is not stable, and the estimated standard deviation is likely to be underestimated compared with the true standard deviation of the model internal variability. Large differences in the mean and a SNR >>1 are, thus, not surprising when analysing decadal periods.
On longer timescales, the estimate of the mean and standard deviation converge toward their `true' values. Accordingly, we see that the differences in the mean between PI_MO and PI_AR become smaller and approach zero (Figure 4d to 9d).
When considering the 200-year long-term mean, we find no SNR value greater than one (Figure 2 and 3). Following this diagnostic, and for the variables we assessed, our results show that there is no significant difference in the simulated mean between the two PI_MO and PI_AR HadGEM3-GC3.1 simulations when considering a 200-year long period.

6) Page 8, lines 16-17: I reject your conclusion that "simulations using the HadGEM3-GC3.1 model are reproducible […] long-enough simulation in length is used". You can only conclude from your analyses that the mean of the variables assessed is not different between MO and AR for your piControl simulations. Actually showing that the model is reproducible would require that your diagnostics provide an exhaustive description of the model physical behaviour, not only mean, but also variability, teleconnection patterns, trends... You can say that your analyses do not show that the model is not reproducible, and that's already a valuable information, that has not been provided by all the modelling centres (and you should receive credit for this).

We take the opportunity to stress that in the paper we present mean and standard deviation (Table 2) for the considered variables, and that teleconnection patterns are indeed discussed, when results point towards a probable change in their mean characteristics (see section 4.2 about the 100-year timescale). However, as more could be done to fully characterize the physical behaviour of the model, we reformulated our sentence at page 8 as follows:

We thus conclude that the mean climate properties simulated by the HadGEM3-GC3.1 model are reproducible on different HPC platforms, provided that a long-enough simulation length is used.

7) Page 8, line 22: I propose "The large differences observed on time-scales shorter than 200-years are a direct consequence of the (potentially underestimated) internal variability of the model, triggered (at least initially) by the machine-dependent processes (compiler […] 3.1 for details)."

The sentence at page 8, line 22, was modified to take into account the reviewer's suggestion:

The large differences observed on time-scales shorter than 200-years are a direct consequence of the (potentially underestimated) internal variability of the model, and triggered (at least initially) by machine-dependent processes (compiler, machine architecture etc., see section 2 and 3.1 for details).

. 8) Page 8, line 27: this is a major point: because the analyses presented in your manuscript concern piControl runs performed with a fully-coupled GCM (typically one of the DECK experiments, see https://www.geosci-model-dev.net/9/1937/2016/gmd-9-1937-2016.pdf), I assume that your are talking about the minimum length of the piControl run in the DECK (if not, then you really need to add more details to be more specific). The actual minimum length for a piControl run in the CMIP6 DECK is 500 years. Therefore your section starts with a wrong statement, which is a pretty serious mistake. Indeed, your results suggest that 100 years may not be enough to fully sample the internal variability of HadGEM3-GC3.1. The good news regarding this statement is that the CMIP6 protocol asked for 500 years. The bad news is that several CMIP6 models have much more internal variability than their previous CMIP5 versions, and that 500 years might not be enough. But this is another story. Also, your conclusions and suggestions have a priori no reason to be applicable to other experiments/MIPs, such as AMIP, historical runs, scenarios, etc... if you want to make on any other given experiment/MIP, be more specific.

In this paragraph we refer to individual Model Intercomparison projects (MIPs), and not to DECK (which the PI control run belongs to) or Historical simulations. This terminology is confirmed by the same reference the reviewer provides in their comment, where the structure of CMIP6 (DECK + Historical + MIPs) is presented and explained.

The minimum simulation length required by many MIPs is 100 years or less. At page 8, line 29-32, we have now added a list of individual MIPs (with references) that recommend 100 years or less, and added a sentence to clarify that we do not refer to DECK or Historical simulations (see response to referee n2, point 1). In relation to the above, please see also the HighMIP documentation paper (https://www.geosci-model-dev.net/9/4185/2016/gmd-9-4185-2016.pdf ) where they say: *"The future end-date is based on a compromise between what is computationally affordable by a sufficient number of centres (~100 years of integration) and what is scientifically relevant."*

The relevance of our results to MIPs is explained in Introduction (p1, lines 17-20) and in the Conclusions (p10, lines 26-31). As an example, two of the authors of the present manuscript are part of the PMIP4 community, the analysis this study is based on was of crucial importance to the UK PMIP community in order to decide how long Tier 1 and 2 experiments should be.

In the manuscript, it is not stated that PI control simulations should be run for longer than 100 years,

but is shown that, for comparison purposes (with the PI control run), other MIPs experiments should be run for at least 200 years when possible. This will assure that the HadGEM3 internal variability is sampled correctly, and that differences in means due to a wrong sampling will not be confused with system responses to a different climate forcing. The paragraph at page 10, lines 26-31, is reformulated to help clarify:

This result has immediate implications for those members of the UK CMIP6 community who will run individual MIP experiments on the ARCHER HPC platform, and will compare results against the reference PI simulation run on the MO platform by the UK Met Office. The magnitude of (PI_MO - PI_AR) differences presented in this paper should be regarded as threshold values below which differences between ARCHER and MO simulations must be interpreted with caution (as they might be the consequence of a wrong sampling of the model internal variability rather than the climate response to a different forcing).

. 9) Page 8, lines 4-14: I don't understand where you want to go with the 2/3 power law, although the result is surprisingly consistent among the variables. And the "plateau" you describe is supported by three consecutive points on your plots on figure 10, the last one being slightly higher than expected by the line. I would agree that there is a plateau if it was described by more than one single point being higher than expected. And once more, you conclude that this results "suggests" something. I would recommend using your results to "show" things, and stick to what they actually show.

We agree that longer simulations, resulting in more data points in Figure 10 and Figure 4d to 9d, would allow a better visualization of the plateau. However, we show that, as the time-scale increases, data points in Figure 10 vary following a power law relationship. (PI_MO – PI_AR) differences are very small and close to zero at the 200-year timescale (Figure 4d to 9d). Since they are expected to become even smaller (closer and closer to zero) with time, the trend exhibited by the data is to eventually plateau at zero for timescales $\geq$ 200 years.

These results are not described as "the data plateaus at the 200-year timescale", rather as "approaches a plateau near the 200-year time-scale" (page 8, line 9). This reflects the reviewer's comment regarding just one point being above the line in Figure 10.

Figure 10 provides an alternative/additional way to quantify the HadGEM3 model behaviour on the two HPC platforms: this analysis tells us that, not only the considered variables converge to their true value at the 200-year timescale (Figure 4d – 9d), but that the rate at which this happens is the same for all variables at the global scale (which is not immediate when you look at Figure 4d - 9d). As the reviewer recognized, this behaviour is remarkably consistent among all the considered variables and is thus worth mentioning.

See below changes at page 8, lines 7-11, and page 10, lines 12-14, to take into account the reviewer's comment:

Thus, the straight-lines that best fit the global mean data in Figure 10 have a slope of ~ 2/3. The existence of a ~ 2/3 power law, which does not depend on the single quantity, shows a consistent

scaling of PI_MO - PI_AR) differences with the timescale that approaches a plateau near the 200-year timescale (note that an actual plateau can only be reached for longer simulations, as differences computed over all timescales longer than 200 years would be ~ 0).

There exist a consistent time-dependent scaling of (PI_MO - PI_AR) differences across the whole climate simulation, so that variables converge toward their true values at the same rate, independently on the physical processes that they represent.

**Mathematical demonstration of $STDEV_X = STDEV_{X-Y} / \sqrt{2}$**

We use the basic property of variance for which:

$$Var_{X-Y} = Var_X + Var_Y - 2Cov(X,Y)$$

If variables are uncorrelated (i.e. independent), so that $ov(X,Y) = 0$ , and have the same variance ($Var_X = Var_Y$) :

$$Var_{X-Y} = 2Var_X$$

As the square root of the variance is the standard deviation:

$$STDEV_{X-Y} = \sqrt{2}\, STDEV_X \quad \rightarrow \quad STDEV_X = STDEV_{X-Y}/\sqrt{2}$$

[revised manuscript text omitted]